# Λ-DARTS: Mitigating Performance Collapse by Harmonizing Operation Selection among Cells

**Sajad Movahedi**[1], **Melika Adabinejad**[1], **Ayyoob Imani**[2], **Arezou Keshavarz**[1], **Mostafa Dehghani**[3]
**Azadeh Shakery**[1,4] **and Babak N. Araabi**[1]
[1]University of Tehran, [2]LMU Munich, [3]Google Brain,
[4] Institute for Research in Fundamental Sciences (IPM)
`{s.movahedi, melika.adabi}@ut.ac.ir, ayyoob.imani@cis.lmu.de`
`arezou@keshavarz.net, dehghani@google.com, {shakery, araabi}@ut.ac.ir`

## Abstract

Differentiable neural architecture search (DARTS) is a popular method for neural architecture search (NAS), which performs cell-search and utilizes continuous relaxation to improve the search efficiency via gradient-based optimization. The main shortcoming of DARTS is performance collapse, where the discovered architecture suffers from a pattern of declining quality during search. Performance collapse has become an important topic of research, with many methods trying to solve the issue through either regularization or fundamental changes to DARTS. However, the weight-sharing framework used for cell-search in DARTS and the convergence of architecture parameters has not been analyzed yet. In this paper, we provide a thorough and novel theoretical and empirical analysis on DARTS and its point of convergence. We show that DARTS suffers from a specific structural flaw due to its weight-sharing framework that limits the convergence of DARTS to saturation points of the softmax function. This point of convergence gives an unfair advantage to layers closer to the output in choosing the optimal architecture, causing performance collapse. We then propose two new regularization terms that aim to prevent performance collapse by harmonizing operation selection via aligning gradients of layers. Experimental results on six different search spaces and three different datasets show that our method (Λ-DARTS) does indeed prevent performance collapse, providing justification for our theoretical analysis and the proposed remedy. We have published our code at `https://github.com/dr-faustus/Lambda-DARTS`.

## 1 Introduction

With the growth of the popularity of deep learning models, neural architecture design has become one of the most important challenges of machine learning. Neural architecture search (NAS), a now prominent branch of AutoML, aims to perform neural architecture design in an automatic way (Elsken et al., 2019; Ren et al., 2021). Initially, the problem of NAS was addressed through reinforcement learning or evolutionary algorithms by the seminal works of (Zoph & Le, 2017; Baker et al., 2017; Stanley & Miikkulainen, 2002; Real et al., 2017). But despite the recent advancements in efficiency (Baker et al., 2018; Liu et al., 2018; Cai et al., 2018), these methods remain impractical and not widely accessible. One-shot methods aim to address this impracticality by performing the architecture search in a one-shot and end-to-end manner (Liu et al., 2019; Guo et al., 2020; Pham et al., 2018; Brock et al., 2018; Bender et al., 2018). Another technique for increasing the efficiency of search is cell-search (Zoph et al., 2018), which performs NAS for a set of cells stacked on top of each other according to a pre-defined macro-architecture. Differentiable neural architecture search (DARTS) (Liu et al., 2019) is a one-shot method that performs cell-search using gradient descent and continuous relaxation. It performs the cell-search using a weight-sharing framework. As a result of these innovations, DARTS is one of the most efficient methods of architecture search (Elsken et al., 2019).

One of the most severe issues of DARTS is the performance collapse problem (Zela et al., 2020), which states that the quality of the architecture selected by DARTS gradually degenerates into an architecture solely consisting of skip-connections. This issue is usually attributed to gradient vanishing (Zhou et al., 2020) or large Hessians of the loss function (Zela et al., 2020), and mostly addressed by heavily modifying DARTS (Chu et al., 2021; Wang et al., 2021; Gu et al., 2021; Ye et al., 2022).

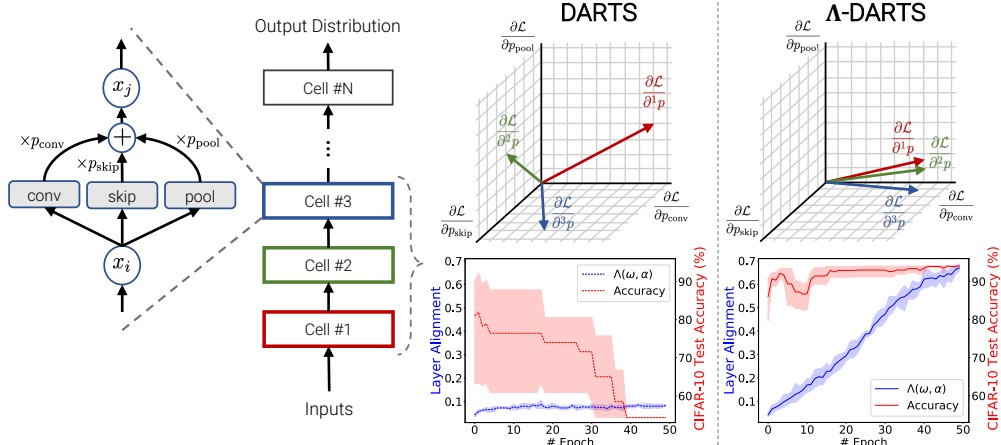

Figure 1: An illustration of the differentiable formulation for NAS by DARTS, and the effects of layer alignment over the trajectory of the performance of the discovered architectures by DARTS and $\Lambda$-DARTS. The experiments are performed on the NAS-Bench-201 search space, averaged over 4 runs with a $95\%$ confidence interval.

In this paper, we argue that these theories fail to grasp the main reason behind the performance collapse: the conditions imposed on the convergence of DARTS by the weight-sharing framework. Specifically, we first define a measure to show the correlation between the gradient of each layer corresponding to the architecture parameters - dubbed *layer alignment* ($\Lambda$). Then, following a careful analysis of the convergence conditions of DARTS and the effects of weight-sharing, we find that:

1. Due to the value of $\Lambda$, DARTS can only achieve convergence by reaching the saturation point of the softmax function, where the normalized architecture parameters become almost one-hot vectors. This convergence point does not depend directly on the loss function, giving an unfair advantage to the layers that do not suffer from gradient vanishing.

2. Low value for $\Lambda$ means that the optimal architecture corresponding to each layer varies wildly, with layers that are closer to the output - and therefore not suffering from gradient vanishing - mainly preferring non-parametric operations.

3. The aforementioned issues are direct contributors to the performance collapse problem. Furthermore, increasing the depth of the search model or the number of iterations used for the search increases the severity of performance collapse.

In Figure-1, we can see an illustration of DARTS and the concept of layer alignment (which we will define in Section-3.2), and observe a clear correlation between the layer alignment and the performance of the discovered architecture on the NAS-Bench-201 search space (Dong & Yang, 2020). In this paper, we will use a combination of analytical and empirical evidence to support the claim that this relationship is in fact a causal relationship, resulting from the weight-sharing framework used in DARTS.

Following these conclusions, we introduce a regularization term that aims to alleviate the problem of performance collapse by increasing the layer alignment. Then, through a comprehensive empirical examination, we show that our method - dubbed $\Lambda$-DARTS - significantly improves the performance of DARTS on various search spaces and datasets without any modification to the algorithm of DARTS or the structure of the cells. Specifically, our method achieves an average accuracy of $96.57\%$ and $83.85\%$ on the CIFAR-10 and CIFAR-100 datasets on the DARTS search space, improving upon the current state-of-the-art by $0.06\%$ and $0.39\%$, respectively. To the best of our knowledge, our work is the first to investigate DARTS from the point-of-view of its *wight-sharing framework* and *convergence conditions*.

## 2 RELATED WORK

DARTS (Liu et al., 2019) proposed a continuous and differentiable search space through weighting a fixed set of operations to make NAS more scalable. It trains a super-graph with gradient descent and chooses the sub-graph consisted of weightiest operation edges. Its simplicity made it very popular and many variations emerged to address its theoretical and empirical setbacks:

**Overlooked effects of design decisions in DARTS.** (Wang et al., 2021) and (Gu et al., 2021) claim that DARTS chooses the final architecture with regard to operation weights, whereas these magnitudes do not necessarily correlate with the performance. (Wang et al., 2021) tries to measure operation impact directly and DOTS (Gu et al., 2021) tries to decouple topology from operation during search. IDARTS (Xue et al., 2021) posits that the innate relationship between the architecture's parameters is ignored by the gradient descent method used in DARTS and formulates their optimization as a bi-linear problem.

**Generalization ability.** The differentiable NAS methods are often executed on a smaller dataset of a certain task to reduce the required computational resources. AdaptNAS (Li et al., 2020), MixSearch (Liu et al., 2021), P-DARTS (Chen et al., 2021b), and DrNAS (Chen et al., 2021a) show that the discovered architecture does not always perform as well as on the proxy task on another more challenging dataset. They solve this issue by providing the network with domain information, making the search network progressively deeper, and reformulating the search algorithm.

**Performance collapse.** Zela et al. (2020); Chu et al. (2021) point out the performance collapse problem of DARTS. They show that DARTS consistently discovers networks with mainly skip-connection operations, which causes severe performance declination. Zela et al. (2020) monitors the eigenvalues of the Hessian of the architecture parameters and performs early stopping. Based on their work, SmoothDARTS (Chen & Hsieh, 2020) performs Hessian regularization to smooth the architecture Hessian. PC-DARTS (Xu et al., 2020) reduces the number of output channels of each operation, for the sake of lower memory consumption and a larger batch size during search. iDARTS (Zhang et al., 2021) focuses on improving the optimization of the architecture parameters by estimating the descent direction more accurately. DARTS- (Chu et al., 2021) argues that skip-connection operations have a potentially sharp loss curvature that leads to their advantage during search, and tries to solve it by adding auxiliary skip-connections to cell structure. $\beta$-DARTS (Ye et al., 2022) replaces the $\ell^2$-regularization on the architecture parameters with a new regularization term to alleviate the unfair advantage of skip-connections. In this paper, we provide a novel analysis of DARTS from the point-of-view of its convergence, diagnosing an overlooked issue of its weight-sharing system that sends convergence down an irrelevant path and may be the root cause of performance collapse. We propose a solution for the matter without modifying the core formulation of DARTS.

## 3 An Analysis on the Convergence of DARTS

We start with an analysis of the weight-sharing framework used for cell-search in DARTS, and the required conditions for its convergence. We then empirically investigate whether DARTS can satisfy any of these conditions, and show that it is only capable of reaching convergence under one of them. Finally, using these observations, we will show that this type of convergence causes performance collapse, providing a motivation for $\Lambda$-DARTS. For the uninitiated reader, we provide a brief explanation of DARTS and the weight-sharing framework used by it for cell-search in Appendix-A.1.

### 3.1 The convergence of DARTS

Assuming the primary loss function $\mathcal{L}(.,.)$ to be differentiable, we can write the necessary and sufficient condition for optimality of the architecture parameters, ($\alpha$) as (Boyd & Vandenberghe, 2014):

$$\nabla_\alpha \mathcal{L}^{val}(\omega, \alpha^*) = 0, \tag{1}$$

where $\omega$ represents the parameters corresponding to the operations, and $\mathcal{L}^{val}(.,.)$ is the loss function calculated over the validation set. Let $\boldsymbol{p} = \sigma(\alpha)$ be the softmax normalized architecture parameters, where $\sigma(.)$ corresponds to the softmax function. Then we can re-write (1) as:

$$\boldsymbol{J}_\sigma(\alpha^*) \nabla_{\boldsymbol{p}} \mathcal{L}(\omega, \alpha^*) = 0, \tag{2}$$

where $\boldsymbol{J}_\sigma(\alpha) = \text{diag}(\sigma(\alpha)) - \sigma(\alpha)\sigma(\alpha)^T$ corresponds to the Jacobian of the softmax function on all edges, and $\text{diag}(.)$ being the diagonal matrix. Note that given the weight-sharing framework, we can write $\nabla_{\boldsymbol{p}} \mathcal{L}(\omega, \alpha^*) = \frac{\partial \mathcal{L}}{\partial p}(\omega, \alpha^*)$ as the sum of gradients received by each layer:

$$\boldsymbol{J}_\sigma(\alpha^*) \left( \sum_{\ell=1}^{L} \nabla_{\ell_{\boldsymbol{p}}} \mathcal{L}(\omega, \alpha^*) \right) = 0. \tag{3}$$

Now note that $\boldsymbol{J}_\sigma(.)$ is a block-diagonal matrix, with each block corresponding to one of the edges in the architecture. Therefore, $\boldsymbol{J}_\sigma(.)$ is a positive semi-definite matrix with its null-space containing

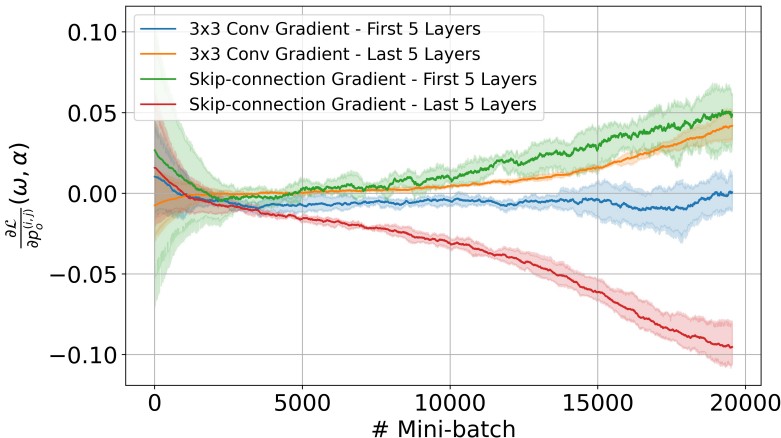

Figure 2: The exponential average (with decay rate $0.999$) of the sum of the gradients of the $3 \times 3$ convolution and skip-connection operations for the first 5 layers and the last 5 layers averaged over all 6 edges of the cell on the NAS-Bench-201 search space, averaged over 4 runs with a $95\%$ confidence interval. Note that negative values in the gradient mean the value of the parameter corresponding to the operation will be increasing.

vectors of the following form:

$$\boldsymbol{v} = \begin{bmatrix} \boldsymbol{v}_1^T & \boldsymbol{v}_2^T & ... & \boldsymbol{v}_{|E|}^T \end{bmatrix}^T, \tag{4}$$

where the vector $\boldsymbol{v}$ consists of concatenated vectors $\boldsymbol{v}_i \in \mathbb{R}^{|E|}$ of the form $\boldsymbol{v}_i^T \in \{c_i \cdot \begin{bmatrix} 1 & 1 & ... & 1 \end{bmatrix}^T \mid c_i \in \mathbb{R}\}$ (Gao & Pavel, 2017). Considering the diverse set of operations used in DARTS, it is not an unfair assumption that $\nabla_{\boldsymbol{p}} \mathcal{L}(\omega, \alpha)$ won't be in the form presented in (4). In fact, we have seen empirical evidence to support this claim, which will be discussed in Section-3.2. So assuming that $\nabla_{\boldsymbol{p}} \mathcal{L}(\omega, \alpha)$ does not belong to the null space of $\boldsymbol{J}_\sigma(\alpha)$, satisfying (1) can only be achieved in two ways:

1. $\nabla_{\boldsymbol{p}} \mathcal{L}(\omega, \alpha^*)$ approaching $0$.
2. The Jacobian $\boldsymbol{J}_\sigma(\alpha^*)$ approaching $0$.

## 3.2 THE LAYER ALIGNMENT ISSUE

Assuming that $\forall \ell: \quad \nabla_{\ell \boldsymbol{p}} \mathcal{L}(\omega, \alpha^*) \neq 0$, a necessary condition for $\nabla_{\boldsymbol{p}} \mathcal{L}(\omega, \alpha^*)$ to be able to approach zero is that for each $\ell$ we have $\exists \ell' \neq \ell: \quad \nabla_{\ell \boldsymbol{p}} \mathcal{L}(\omega, \alpha^*) \not\parallel \nabla_{\ell' \boldsymbol{p}} \mathcal{L}(\omega, \alpha^*)$. Because otherwise, we can write $\nabla_{\boldsymbol{p}} \mathcal{L}(\omega, \alpha^*)^T \nabla_{\ell \boldsymbol{p}} \mathcal{L}(\omega, \alpha^*) = \|\nabla_{\ell \boldsymbol{p}} \mathcal{L}(\omega, \alpha^*)\|_2^2 \neq 0$. Empirically, we noticed that this is not the case in DARTS. More specifically, we define the function $\Lambda$ - which we dub *layer alignment* - as:

$$\Lambda(\omega, \alpha) = \frac{1}{\binom{L}{2}} \sum_{\ell < \ell'} \frac{\nabla_{\ell \boldsymbol{p}} \mathcal{L}(\omega, \alpha)^T \nabla_{\ell' \boldsymbol{p}} \mathcal{L}(\omega, \alpha)}{\|\nabla_{\ell \boldsymbol{p}} \mathcal{L}(\omega, \alpha)\|_2 \|\nabla_{\ell' \boldsymbol{p}} \mathcal{L}(\omega, \alpha)\|_2}. \tag{5}$$

We noticed that regardless of the search space, the layer alignment is always near zero, i.e. the layer gradients are almost orthogonal. In Figure 1, we can see the value of $\Lambda(.,.)$ on the NAS-Bench-201 search space (Dong & Yang, 2020) for both DARTS and $\Lambda$-DARTS. In Appendix-A.2, we try to detect the reason behind the low layer alignment issue.

As a result of this orthogonality, we expect that under the aforementioned assumptions DARTS is incapable of reaching convergence the first way, since it is likely incapable of satisfying the necessary condition for it. So we will propose a lower-bound on the value $\|\nabla_\alpha \mathcal{L}(\omega, \alpha)\|_2^2$, which formalizes our claim:

**Proposition 1** *Let $\nabla_\alpha \mathcal{L}(\omega, \alpha)$ be orthogonal to the null space of $\boldsymbol{J}_\sigma(\alpha)$, and assuming we have $L$ layers of cells in our search network, we have $\forall \ell \neq \ell' \quad \nabla_{\ell \boldsymbol{p}} \mathcal{L}(\omega, \alpha) \perp \nabla_{\ell' \boldsymbol{p}} \mathcal{L}(\omega, \alpha)$. Let $\lambda_i(\alpha) > 0$ be the set of eigenvalues corresponding to $\boldsymbol{J}_\sigma(\alpha)$. Then the square norm of the gradient $\nabla_\alpha \mathcal{L}(\omega, \alpha)$ can be bounded by:*

$$\|\nabla_\alpha \mathcal{L}(\omega, \alpha)\|_2^2 \geq \min_{i, \lambda_i(\alpha) \neq 0} \lambda_i(\alpha)^2 \cdot L \cdot \min_\ell \|\nabla_{\ell \boldsymbol{p}} \mathcal{L}(\omega, \alpha)\|_2^2. \tag{6}$$

We provide a proof for this proposition in Appendix-A.10. This proposition shows that for any value of $\min_\ell \|\nabla_{\ell_p}\mathcal{L}(\omega,\alpha)\|_2^2$, we have a large enough value for $L$ that prevents the right-hand-side of (6) to become too small. As a result, for a deep enough search network, the only way for $\alpha$ to reach convergence is through reaching the softmax saturation point, wherein we have near zero eigenvectors corresponding to $J_\sigma(\alpha)$, and the normalized architecture parameters become close to one-hot vectors. This phenomenon has been observed in the literature (Bi et al., 2019; Dong & Yang, 2020), where DARTS usually converges to a softmax saturation point with all of the edges selecting the skip-connection or none operations, depending on the search space. In Figure-3, we can see this effect, where the absolute changes in the softmax normalized architecture weights of DARTS during search is rapidly growing, compared to Λ-DARTS which reaches a plateau during search. Note that converging to a softmax saturation point is not an inherently bad outcome, because it corresponds to near zero optimality gap when using continuous relaxation (Boyd & Vandenberghe, 2014). But as we will see, this characteristic gives an advantage to layers closer to the output in selecting the optimal architecture, contributing to performance collapse (Zela et al., 2020).

### 3.3 THE PERFORMANCE COLLAPSE

Another issue caused by low layer alignment is that the optimal architectures corresponding to each layer are going to be vastly different. In Figure 2 we can see the exponential average of the sum of the gradients for convolution and skip-connection operations in the first 5 layers (furthest from the output of the network) and the last 5 layers (closest to the output of the network) averaged over all edges in a search cell on the NAS-Bench-201 search space (Dong & Yang, 2020). As evident, the optimal cell corresponding to the shallower layers (closer to the input) mostly contains convolution operations, while the optimal cell corresponding to the deeper layers (closer to the output) mostly contains skip-connection operations. We attribute this observation to skip-connections having a proclivity towards allowing the gradients to flow further into the network, while convolutions provide more complex features for upper layers. Furthermore, due to gradient vanishing, the gradients corresponding to the skip-connection operations on average have larger magnitudes, with the magnitude of the layer gradients gradually becoming smaller the deeper we go. As a result, $\nabla_\alpha\mathcal{L}(\omega,\alpha)$ will have much smaller negative values corresponding to the skip-connection operations.

Using these observations, we can clearly bridge the analytical gap between the performance collapse problem and the layer alignment. Due to the weight-sharing structure, the value of the function $\Lambda(\omega,\alpha)$ is close to zero. As a result, DARTS can only achieve convergence in the softmax saturation points. Furthermore, given the low layer alignment, the optimal architecture from the perspective of the layers are going to be extremely varied, with deeper layers having a strong inclination towards skip-connections. This gives skip-connections a clear advantage due to the larger magnitude of their gradients. So as the search progresses, the architecture parameters corresponding to the skip-connection operations become larger than the convolution operations, eventually resulting in a saturation point that overwhelmingly selects skip-connection operations (Bi et al., 2019; Chu et al., 2021). As the depth of the search network increases, vanishing

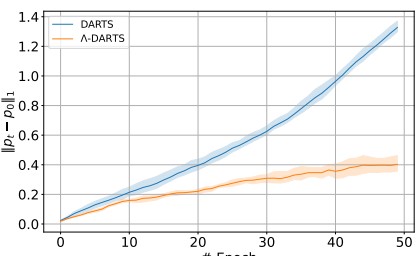

Figure 3: The $\ell^1$ norm of changes in softmax-normalized architecture weights, averaged over 4 runs with a $95\%$ confidence interval. The search is performed on CIFAR-10.

gradient becomes more severe, further exacerbating the issue (He et al., 2016). This is evident when comparing the results of DARTS on its original search space and the NAS-Bench-201 search space, which utilizes deeper search networks (Dong & Yang, 2020). Similarly, increasing the number of iterations used for the search results in further exacerbation of the performance collapse problem (Bi et al., 2019), which can be attributed to the severity of vanishing gradient increasing as the loss function approaches its optimal point (He et al., 2016). These observations motivate us to alleviate both of these issues caused by low layer alignment through increasing the value of $\Lambda(\omega,\alpha)$. In Appendix-A.3 we provide a thorough analysis that connects our method to some of the prior works.

## 4 THE PROPOSED METHOD TO MITIGATE PERFORMANCE COLLAPSE

In the previous segment, we saw that the main contributor to the performance collapse problem is low layer alignment (i.e. $\Lambda(\omega,\alpha)\approx0$). Now we will introduce two regularization terms based on this value to solve

performance collapse. Furthermore, we will introduce a method to estimate the gradient corresponding to the regularization terms that grow linearly with the size of the network.

## 4.1 THE REGULARIZATION TERM

For the sake of simplicity, let's assume we aim to search for a single cell. To increase the value of $\Lambda(\omega,\alpha)$, we can directly use the function itself, since it is a differentiable function. So we define our main regularization term as the function $\Lambda(\omega,\alpha)$, which we aim to maximize.

In our preliminary experiments, we noticed that in case the dimension of $\alpha$ is large, $\Lambda(\omega,\alpha)$ only concentrates on the operations with relatively larger gradients. This means that while the angle between the gradients of the layers is reduced, only the gradient of a small portion of the operations will have the same sign, resulting in lower performance. So given $^\ell g = \nabla_{\ell_{\boldsymbol{p}}}\mathcal{L}(\omega,\alpha)$ and $^{\ell'}g = \nabla_{\ell'_{\boldsymbol{p}}}\mathcal{L}(\omega,\alpha)$, we will also introduce another regularization term that aims to increase the correlation between the signs of the gradients of each layer:

$$\Lambda_{\pm}(\omega,\alpha) = \frac{1}{\binom{L}{2}}\sum_{\ell<\ell'}\frac{^\ell g^T {}^{\ell'}g}{|^\ell g|^T |^{\ell'}g|},\tag{7}$$

where $|.|$ corresponds to the element-wise absolute value function. Note that while the optimal point of $\Lambda(\omega,\alpha)$ corresponds to $\nabla_{\ell_{\boldsymbol{p}}}\mathcal{L}(\omega,\alpha) = c\nabla_{\ell'_{\boldsymbol{p}}}\mathcal{L}(\omega,\alpha)$, where $c$ is a constant, the optimal point of $\Lambda_{\pm}(\omega,\alpha)$ corresponds to $\text{sign}(\nabla_{\ell_{\boldsymbol{p}}}\mathcal{L}(\omega,\alpha)) = \text{sign}(\nabla_{\ell'_{\boldsymbol{p}}}\mathcal{L}(\omega,\alpha))$, where $\text{sign}(.)$ is the element-wise signum function.

Given one of the regularization terms above, the bi-level optimization problem of $\Lambda$-DARTS will become:

$$\begin{aligned}\min_{\alpha}\quad & \mathcal{L}^{val}(\omega^*(\alpha),\alpha)\\ \text{s.t.}\quad & \omega^*(\alpha) = \operatorname*{argmin}_{\omega}\mathcal{L}^{train}(\omega,\alpha) - \lambda\Lambda^{train}(\omega,\alpha),\end{aligned}\tag{8}$$

where we insert the regularization term in the inner objective. In Appendix-A.4, we show that optimizing the regularization term w.r.t. the outer objective performs poorly. Similar to most variants of DARTS, we use a linear scheduling for $\lambda$ (Zela et al., 2020; Chen & Hsieh, 2020): i.e. $\lambda_t = \lambda \times \frac{t}{T}$, where $t$ is the number of current epoch and $T$ is the total number of epochs used for search.

## 4.2 ESTIMATING THE GRADIENTS OF THE REGULARIZATION TERMS

Assuming the gradients corresponding to the operations are non-zero, both $\Lambda(\omega,\alpha)$ and $\Lambda_{\pm}(\omega,\alpha)$ are differentiable. So their gradient with respect to $\omega$ can be written as:

$$\nabla_{\omega}\Lambda(\omega,\alpha) = \frac{1}{\binom{L}{2}}\sum_{\ell=1}^{L}\nabla^2_{\omega,\ell_p}\mathcal{L}(\omega,\alpha)\sum_{\ell'\neq\ell}\delta^{\ell,\ell'},\tag{9}$$

where we have:

$$\begin{aligned}\delta_{\Lambda}^{\ell,\ell'} &= \left(\boldsymbol{I} - \frac{^\ell g\, {}^\ell g^T}{\|^\ell g\|_2^2}\right)\frac{^{\ell'}g}{\|^\ell g\|_2\|^{\ell'}g\|_2},\\ \delta_{\Lambda_{\pm}}^{\ell,\ell'} &= \left(\boldsymbol{I} - \frac{^\ell g^T {}^{\ell'}g}{|^\ell g|^T |^{\ell'}g|}\text{diag}(\text{sign}(^\ell g))\text{diag}(\text{sign}(^{\ell'}g))\right)\frac{^{\ell'}g}{|^\ell g|^T |^{\ell'}g|},\end{aligned}\tag{10}$$

with $\boldsymbol{I}$ corresponding to the identity matrix. Note that calculating $\nabla_{\omega}\Lambda(\omega,\alpha)$ will be of $O(L\cdot|\omega|\cdot|\alpha|)$ due to the existence of second derivative matrices for each layer. So a direct calculation of (9) is out of the question. But similar to (Liu et al., 2019), we can use finite difference approximation to estimate $\nabla_{\omega}\Lambda(\omega,\alpha)$.

Let $\mathcal{L}(\omega,\boldsymbol{P})$ be the loss function as a function of $\omega$ and $\boldsymbol{P}\in\mathbb{R}^{L\times|\alpha|}$, where:

$$\boldsymbol{P} = \begin{bmatrix}\sigma(\alpha) & \sigma(\alpha) & ... & \sigma(\alpha)\end{bmatrix}^T\tag{11}$$

corresponds to the softmax normalized architecture weights used for each layer in the search network. Furthermore, let $\Delta\in\mathbb{R}^{L\times|\alpha|}$ be:

$$\Delta = \begin{bmatrix}\sum_{\ell'\neq 1}\delta^{1,\ell'} & \sum_{\ell'\neq 2}\delta^{2,\ell'} & ... & \sum_{\ell'\neq L}\delta^{L,\ell'}\end{bmatrix}^T.\tag{12}$$

Table 1: Comparison with state-of-the-art on the NAS-Bench-201 search space. Mean and standard deviation of the accuracy were calculated over 4 independent runs. The search is performed on the CIFAR-10 dataset. The best performance is boldface. (1st): first-order, (2nd): second-order

| Method | CIFAR-10 | | CIFAR-100 | | ImageNet16-120 | |
|---|---|---|---|---|---|---|
| | Valid | Test | Valid | Test | Valid | Test |
| DARTS (1st) (Liu et al., 2019) | 39.77±0.00 | 54.30±0.00 | 15.03±0.00 | 15.61±0.00 | 16.43±0.00 | 16.32±0.00 |
| DARTS (2nd) (Liu et al., 2019) | 39.77±0.00 | 54.30±0.00 | 15.03±0.00 | 15.61±0.00 | 16.43±0.00 | 16.32±0.00 |
| GDAS (Dong & Yang, 2019) | 89.89±0.08 | 93.61±0.09 | 71.34±0.04 | 70.70±0.30 | 41.59±1.33 | 41.71±0.98 |
| SNAS (Xie et al., 2019) | 90.10±1.04 | 92.77±0.83 | 69.69±2.39 | 69.34±1.98 | 42.84±1.79 | 43.16±2.64 |
| DSNAS (Hu et al., 2020) | 89.66±0.29 | 93.08±0.13 | 30.87±16.40 | 31.01±16.38 | 40.61±0.09 | 41.07±0.09 |
| PC-DARTS (Xu et al., 2020) | 89.96±0.15 | 93.41±0.30 | 67.12±0.39 | 67.48±0.89 | 40.83±0.08 | 41.31±0.22 |
| iDARTS (Zhang et al., 2021) | 89.86±0.60 | 93.58±0.32 | 70.57±0.24 | 70.83±0.48 | 40.38±0.59 | 40.89±0.68 |
| DARTS- (Chu et al., 2021) | 91.03±0.44 | 93.80±0.40 | 71.36±1.51 | 71.53±1.51 | 44.87±1.46 | 45.12±0.82 |
| $\beta$-DARTS (Ye et al., 2022) | **91.55±0.00** | **94.36±0.00** | **73.49±0.00** | **73.51±0.00** | **46.37±0.00** | **46.34±0.00** |
| $\Lambda$-DARTS - $\Lambda(\omega,\alpha)$ | **91.55±0.00** | **94.36±0.00** | **73.49±0.00** | **73.51±0.00** | **46.37±0.00** | **46.34±0.00** |
| Optimal (Dong & Yang, 2020) | 91.61 | 94.37 | 73.49 | 73.51 | 46.77 | 47.31 |

Then we can write:

$$\nabla_\omega \Lambda(\omega,\alpha) = \frac{1}{\binom{L}{2}}\left[\frac{\nabla_\omega \mathcal{L}(\omega,\boldsymbol{P}+\epsilon\Delta)-\nabla_\omega \mathcal{L}(\omega,\boldsymbol{P}-\epsilon\Delta)}{2\epsilon}\right]+O(\epsilon^2), \qquad (13)$$

where we set $\epsilon = \frac{\epsilon_0}{\|\Delta\|_F}$. The cost of using this approximation technique is two extra forward-backward pass, which is of $O(|\omega|+|\alpha|)=O(|\omega|)$. So we have eliminated the $L \cdot |\alpha|$ factor from the complexity of $\Lambda$-DARTS, which significantly reduces the computational complexity, while completely eliminating the memory overhead. In Appendix-A.8 we provide the cost of our method in terms of GPU days along with another approximation technique which can reduce the overhead of $\Lambda$-DARTS significantly.

## 5 EXPERIMENTAL RESULTS AND DISCUSSIONS

In this section, we will show the effectiveness of $\Lambda$-DARTS by performing experiments on the NAS-Bench-201 (Dong & Yang, 2020) and DARTS (Liu et al., 2019) search spaces. We will then demonstrate the robustness of $\Lambda$-DARTS by performing more experiments on the reduced DARTS search spaces proposed by (Zela et al., 2020) on different datasets. Finally, by performing an ablation study, we will analyze the effects of the hyper-parameters of $\Lambda$-DARTS over the performance of DARTS. For all experiments, we report the mean and the standard deviation of $\Lambda$-DARTS for 4 independent searches with different random seeds, as well as the result of the best performing architecture, except for the experiments on the reduced DARTS search spaces, wherein only the best performance is reported. In all experiments, unless explicitly mentioned otherwise, the search is performed on CIFAR-10. For details of the experiment settings, we direct the reader to Appendix-A.9. In Appendix-A.11 we provide the architectures discovered by $\Lambda$-DARTS on these search spaces and datasets. For further experiments, we refer the reader to Appendix-A.5, Appendix-A.6, and Appendix-A.7.

### 5.1 NAS-BENCH-201 SEARCH SPACE

In order to demonstrate the effectiveness of $\Lambda$-DARTS in a reproducible setting, we used the NAS-Bench-201 search space for our first set of experiments. In Table-1 we can see the results of $\Lambda$-DARTS compared to other DARTS-based methods. We omit from reporting the results of $\Lambda_\pm(\omega,\alpha)$ here, since it performed similar to $\Lambda(\omega,\alpha)$. As evident, $\Lambda$-DARTS performs exceptionally well compared to the baselines, surpassing all but one of them in terms of the mean and standard deviation of the performance. Compared to the current state-of-the-art (Ye et al., 2022), $\Lambda$-DARTS discovers the same architecture in all 4 experiments. Considering how close to the optimal the performance of $\Lambda$-DARTS is, we attribute the slightly less than optimal performance to the high variance of the CIFAR-10 dataset. Furthermore, we note that the method proposed in (Ye et al., 2022) converges to its optimal architecture very early in the search process (the first 10 epochs), which may indicate a premature convergence to a sub-optimal architecture.

In comparison, $\Lambda$-DARTS does not reach its optimal performance until late in the search procedure, where the performance of the search model is close to optimal and the correlation between the gradient of the layers is very high. In fact, in the DARTS search space we will show further evidence for this claim, where $\Lambda$-DARTS is capable of surpassing the performance of (Ye et al., 2022) on the DARTS search space by a large margin on two datasets. In Figure-1, we can see a clear correlation between the layer alignment and the accuracy of the architecture on CIFAR-10, which reaches its optimal point at around the $40^{th}$

Table 2: Comparison with state-of-the-art on the DARTS search space. The first block contains methods that report the best performing architecture, while the second block contains methods that report the average of multiple searches (except for results on ImageNet, where always the best performance is reported). Our reported mean and standard deviation of the accuracy are calculated over 4 independent runs. We perform the search on the CIFAR-10 dataset. The best performance is boldface on each block. (1st): first-order, (2nd): second-order, (avg): average performance, (best): best performance.
[†] Different search space used for ImageNet.

| Method | CIFAR-10 | | CIFAR-100 | | ImageNet | |
|---|---|---|---|---|---|---|
| | Params (M) | Test Acc (%) | Params (M) | Test Acc (%) | Params (M) | Test Acc (%) |
| NASNet-A (Zoph et al., 2018) | 3.3 | 97.35 | 3.3 | 83.18 | 5.3 | 74.0 |
| DARTS (1st) (Liu et al., 2019) | 3.4 | 97.00±0.14 | 3.4 | 82.46 | - | - |
| DARTS (2nd) (Liu et al., 2019) | 3.3 | 97.24±0.09 | - | - | 4.7 | 73.3 |
| SNAS (Xie et al., 2019) | 2.8 | 97.15±0.02 | 2.8 | 82.45 | 4.3 | 72.7 |
| GDAS (Dong & Yang, 2019) | 3.4 | 97.07 | 3.4 | 81.62 | 5.3 | 74.0 |
| P-DARTS (Liu et al., 2018) | 3.4 | **97.50** | 3.6 | 82.51 | 5.1 | 75.3 |
| PC-DARTS (Xu et al., 2020) | 3.6 | 97.43±0.07 | 3.6 | **83.10** | 5.3 | **75.8** |
| DrNAS (Chen et al., 2021a) | 4.0 | 97.46±0.03 | - | - | 5.2 | **75.8** |
| R-DARTS (Zela et al., 2020) | - | 97.05±0.21 | - | 81.99±0.26 | - | - |
| P-DARTS (Liu et al., 2018) | 3.3±0.21 | 97.19±0.14 | - | - | - | - |
| SDARTS-ADV (Chen & Hsieh, 2020) | 3.3 | 97.39±0.02 | - | - | 5.4 | 74.8 |
| DOTS (Gu et al., 2021) | 3.5 | **97.51±0.06** | 4.1 | **83.52±0.13** | 5.2 | 75.7 |
| DARTS+PT (Wang et al., 2021) | 3.0 | 97.39±0.08 | - | - | 4.6 | 74.5 |
| DARTS- (Chu et al., 2021)[†] | 3.5±0.13 | 97.41±0.08 | 3.4 | 82.49±0.25 | 4.9 | **76.2** |
| β-DARTS (Ye et al., 2022) | 3.8±0.08 | 97.49±0.07 | 3.8±0.08 | 83.48±0.03 | 5.4 | 75.8 |
| Λ-DARTS - $\Lambda(\omega,\alpha)$ (avg) | 3.5±0.13 | 97.48±0.11 | 3.6±0.13 | 83.47±0.20 | - | - |
| Λ-DARTS - $\Lambda(\omega,\alpha)$ (best) | 3.5 | 97.59 | 3.6 | 83.64 | 5.1 | 75.4 |
| Λ-DARTS - $\Lambda_{\pm}(\omega,\alpha)$ (avg) | 3.6±0.13 | **97.57±0.05** | 3.6±0.1 | **83.85±0.38** | - | - |
| Λ-DARTS - $\Lambda_{\pm}(\omega,\alpha)$ (best) | 3.8 | **97.65** | 3.8 | **84.20** | 5.2 | **75.7** |

search epoch. Comparing the results to DARTS, we see that the performance of the discovered architecture declines consistently, eventually reaching an extremely low point.

## 5.2 DARTS SEARCH SPACE

In order to show that Λ-DARTS is effective in mitigating the performance collapse problem in a complex search space, we used the original search space of DARTS proposed in (Liu et al., 2019) for our second experiment. In Table-2, we can see the results of Λ-DARTS compared to the baselines. Λ-DARTS is able to achieve state-of-the-art results on the CIFAR-10 and CIFAR-100 datasets, surpassing the baseline models by a large margin on the more complex dataset of CIFAR-100. Furthermore, Λ-DARTS achieves comparable results on ImageNet, with about 0.1% difference in accuracy compared to the state-of-the-art results on the same macro-architecture. Furthermore, compared to (Ye et al., 2022), we see that Λ-DARTS achieves better performance on two of the three datasets, supporting our claim that β-DARTS may be converging to sub-optimal architectures due to pre-mature convergence. Interestingly, compared to most baselines, Λ-DARTS does not select unnecessarily complex architectures, with the number of parameters averaging around 3.5 on $\Lambda(\omega,\alpha)$ and 3.6 on $\Lambda_{\pm}(\omega,\alpha)$ on CIFAR-10. This means that the architectures are much more efficient in terms of performance, further proving the superiority of Λ-DARTS. This is especially the case on the ImageNet search space, where our model is well within the boundaries of the mobile setting introduced in (Liu et al., 2019).

Comparing $\Lambda(\omega,\alpha)$ with $\Lambda_{\pm}(\omega,\alpha)$, we see that the second function achieves a much better performance. Comparing the discovered cells, we noticed that $\Lambda_{\pm}(\omega,\alpha)$ usually discovers deeper cells with more parametric operations, a quality that highly correlates with the performance of the model. This observation can be attributed to the large dimension of $\alpha$ on this search space.

## 5.3 REDUCED SEARCH SPACES

In order to show the effectiveness of Λ-DARTS in alleviating the performance collapse problem even in the most extreme circumstances, we used the reduced search spaces proposed in (Zela et al., 2020) for further experimentation. In Table-3 we can see the results of Λ-DARTS compared to the baselines. Out of 12 different experiments performed, Λ-DARTS performs better than the baselines in 10 of them, with its performance in the other two experiments being comparable to the baselines. Following these observations, we can claim that our model is robustly stepping out of the performance collapse trap without any modifications to the original formulation of DARTS.

Table 3: Comparison with state-of-the-art on the reduced DARTS search spaces. The test error rate (%) of the best performing architecture from 4 independent runs is reported. The best performance is boldface.

| Dataset | Search Space | DARTS | PC-DARTS | R-DARTS | SDARTS-RS | DARTS+PT | Λ-DARTS |
|---------|--------------|-------|----------|---------|-----------|----------|---------|
| CIFAR-10 | S1 | 3.84 | 3.11 | **2.78** | **2.78** | 3.50 | 2.83 |
| | S2 | 4.85 | 3.02 | 3.31 | 2.75 | 2.79 | **2.56** |
| | S3 | 3.34 | 2.51 | 2.51 | 2.53 | 2.49 | **2.38** |
| | S4 | 7.20 | 3.02 | 3.56 | 2.93 | 2.64 | **2.46** |
| CIFAR-100 | S1 | 29.46 | 24.69 | 24.25 | 23.51 | 24.48 | **22.79** |
| | S2 | 26.05 | 22.48 | 22.44 | 22.28 | 23.16 | **21.68** |
| | S3 | 28.90 | 21.69 | 23.99 | 21.09 | 22.03 | **21.03** |
| | S4 | 22.85 | 21.50 | 21.94 | 21.46 | 20.80 | **20.65** |
| SVHN | S1 | 4.58 | 2.47 | 4.79 | **2.35** | 2.62 | 2.39 |
| | S2 | 3.53 | 2.42 | 2.51 | 2.39 | 2.53 | **2.37** |
| | S3 | 3.41 | 2.41 | 2.48 | 2.36 | 2.42 | **2.31** |
| | S4 | 3.05 | 2.43 | 2.50 | 2.46 | 2.42 | **2.34** |

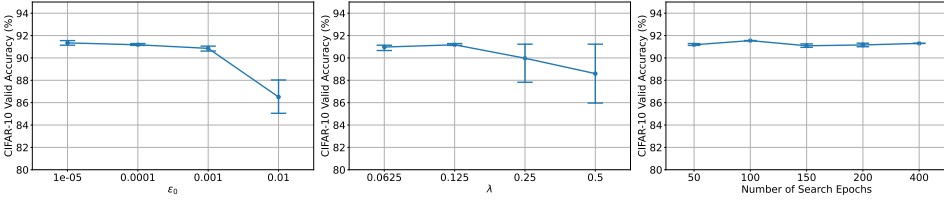

(a) Effects of $\epsilon_0$.      (b) Effects of $\lambda$.      (c) Effects of number of epochs.

Figure 4: An analysis of the effects of hyper-parameters on Λ-DARTS. We report the mean and 95% confidence interval of CIFAR-10 validation accuracy, calculated over 4 runs. (a) Effects of $\epsilon_0$ over the accuracy of the approximation and the performance of discovered architectures. (b) Effects of $\lambda$ over the performance of the discovered architectures. (c) Effects of the number of search epochs on the performance of the discovered architectures. The search is performed on CIFAR-10.

Comparing the baselines, we can see that while DARTS+PT performs better on the S4 search space, SDARTS performs mostly better on the other search spaces. Furthermore, we see similar inconsistencies in PC-DARTS and R-DARTS, where they both perform well on S3 but usually not as well on the other search spaces. On the other hand, Λ-DARTS consistently performs well across all search spaces and datasets, which further shows the robustness of our method.

## 5.4 ABLATION STUDY

In order to assess how the hyper-parameters affect Λ-DARTS, we conducted an ablation study on $\epsilon_0$ and $\lambda$ on the NAS-Bench-201 search space. Furthermore, to show that Λ-DARTS has solved the performance collapse issue, we perform the search for longer epochs (starting from 50, up to 400 epochs) following (Chu et al., 2021; Ye et al., 2022). In Figure 4a, we can see the effect of $\epsilon_0$ over the performance of the discovered architecture. We observe that smaller values of $\epsilon_0$ correlate positively with the quality of the discovered architecture. Considering that lower values of $\epsilon_0$ correspond to lower approximation error in (13), this is expected behavior. But we note that due to the problem of round-off error, we expect this trend to reverse for smaller values of $\epsilon_0$ (Shen et al., 2020). In fact, we can see that the variance of the performance is already starting to increase for the smallest value of $\epsilon_0$.

Figure 4b shows the effect of $\lambda$ over the performance of the discovered architecture. Clearly, the model is not very sensitive to the value of $\lambda$, with the best performing value and the worst performing value having a difference of about 2% in their mean accuracy. Furthermore, the best performing value at $\lambda = 0.125$ corresponds to an average length normalized dot-product of about 0.7, while the worst performing value at $\lambda = 0.5$ is around 0.9. So we are well within the realm of diminishing returns for larger $\lambda$s, and as a result of increased emphasis on the regularization term in the Pareto-optimal point, we observe negative effects from the main objective being sidetracked in the form of lower quality architectures.

We can see the performance of Λ-DARTS for different numbers of search epochs in Figure-4c. Unlike other methods that try to address the issue of performance collapse for larger number of epochs (Bi et al., 2019; Chu et al., 2021), we see a negligible drop in performance (about 0.4%) when comparing the best performing and the worst performing setting. This is a clear indication that the models have achieved convergence at around 100 epochs, reaching a local-minima that corresponds to near-optimal architectures. In fact, we can observe further proof for the convergence of Λ-DARTS in Figure-3, where clearly the absolute changes in the softmax normalized architecture weights has reached a plateau in 50 epochs of search, resulting in a concave curvature.

## 6 CONCLUSION

In this paper, we use a set of analytical and empirical tools to show that the main reason behind performance collapse in DARTS is the lack of correlation between the gradient of its layers, caused by the multi-layer structure and the use of weight-sharing for cell-search. Based on the aforementioned analysis, we propose $\Lambda$-DARTS which manages to achieve better or comparable performance when compared to state-of-the-art baselines. Here, we focused our discussion and experiments on DARTS, however the analysis and ideas are extendable to any other type of differentiable NAS model that performs cell search using weight sharing, which we plan to study in future works. Furthermore, we point out to the cost of estimating the gradient of the regularization terms proposed here as another potential subject for further research.

### ACKNOWLEDGMENTS

We thank colleagues and students at the University of Tehran for all the useful discussions. We would like to also thank researchers at Google for feedback, in particular Hanxiao Liu, Xuanyi Dong, and Avital Oliver for their detailed and valuable comments on the paper. We also thank the reviewers for their constructive and helpful feedback, which helped improving the quality of the paper.

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

# A  APPENDIX

## A.1  REVIEW ON DIFFERENTIABLE NEURAL ARCHITECTURE SEARCH (DARTS)

### A.1.1  CONTINUOUS RELAXATION

For simplicity, let's assume we are searching for a single cell. Let the vector $\alpha \in \mathbb{R}^{|E| \cdot |\mathcal{O}|}$ be the set of parameters corresponding to the architecture, where $E$ is the set of all the possible edges in the computational graph of the cell, and $\mathcal{O}$ is a set of pre-defined operations. The goal is to find the optimal topology and operations in the continuous space using gradient-descent, and then transfer it from the continuous space to the discrete space using the argmax operator to get a near-optimal architecture.

Specifically, let $x_j$ be the output of the $j^{th}$ node in the computational graph; We can write $x_j$ as:

$$x_j = \sum_{i<j} \sum_{o \in \mathcal{O}} p_o^{(i,j)} o(x_i), \tag{14}$$

where $p_o^{(i,j)}$ is the softmax normalized architecture weights: $p_o^{(i,j)} = \frac{\exp(\alpha_o^{(i,j)})}{\sum_{o' \in \mathcal{O}} \exp(\alpha_{o'}^{(i,j)})}$. DARTS then solves the following bi-level optimization problem to find the optimal value for $\alpha$:

$$\begin{aligned} \min_{\alpha} \quad & \mathcal{L}^{val}(\omega^*(\alpha), \alpha) \\ \text{s.t.} \quad & \omega^*(\alpha) = \operatorname*{argmin}_{\omega} \mathcal{L}^{train}(\omega, \alpha), \end{aligned} \tag{15}$$

where $\mathcal{L}^{train}(.,.)$ and $\mathcal{L}^{val}(.,.)$ correspond to the loss function calculated on the train and validation set, and $\omega$ is the set of parameters corresponding to the operations. There are two methods proposed by the original paper to solve the problem above using gradient descent. The first method - dubbed first-order - performs gradient descent on $\alpha$ and $\omega$ in an interleaved fashion, while the second method - dubbed second-order - considers the dependency between $\omega^*(\alpha)$ and $\alpha$ when calculating the descent direction corresponding to $\alpha$. In this paper, we focus on the first-order method since it is more widely used (Elsken et al., 2019). For further information, we direct the reader to the original paper (Liu et al., 2019).

### A.1.2 WEIGHT-SHARING FOR CELL-SEARCH

In order to perform cell-search, DARTS utilizes a weight-sharing framework. Specifically, in a cell-search scheme, $L$ layers of the same cell are stacked on top of each other according to a pre-defined macro architecture. These cells share the same $\alpha$ - i.e. share the same architecture parameter - but have different parameters corresponding to the operations. So they have the same structure, but different operation weights. As a result, we can write the gradient corresponding to the architecture parameter as the sum of the gradients received by each layer.

In order to define this weight-sharing framework concretely, let $^{\ell}x_j$ correspond to the output of the $j^{th}$ node of the architecture graph produced by the $\ell^{th}$ layer. Then we can write the gradient received by $p_o^{(i,j)}$ from the $\ell^{th}$ layer as:

$$\frac{\partial \mathcal{L}}{\partial^{\ell} p_o^{(i,j)}} = \frac{\partial \mathcal{L}}{\partial^{\ell} x_j} \cdot o(^{\ell}x_i). \tag{16}$$

In other words, the gradient received by $p_o^{(i,j)}$ from the $\ell^{th}$ layer corresponds to the dot product of the output of the $i^{th}$ node in $\ell^{th}$ layer and the gradient of the $j^{th}$ node in $\ell^{th}$ layer. Since we have $L$ layers in the macro-architecture, we can write the gradient corresponding to $p_o^{(i,j)}$ as the sum of the gradients received by each layer:

$$\frac{\partial \mathcal{L}}{\partial p_o^{(i,j)}} = \sum_{\ell=1}^{L} \frac{\partial \mathcal{L}}{\partial^{\ell} p_o^{(i,j)}}. \tag{17}$$

Now, by writing the gradient for all edges and operations for $p$ in vector form, we have:

$$\nabla_{\boldsymbol{p}} \mathcal{L}(\omega, \alpha) = \sum_{\ell=1}^{L} \nabla_{\ell \boldsymbol{p}} \mathcal{L}(\omega, \alpha). \tag{18}$$

### A.2 THE ROOT CAUSE OF LOW LAYER ALIGNMENT

An important matter to consider is the root cause of low layer alignment, which can help us further understand the reason behind the performance collapse, and the superior performance of $\Lambda$-DARTS. Here, we will consider two likely candidates that may have caused this issue:

- High level of approximation in the formulation of first-order DARTS (Zhang et al., 2021).
- Learning unrolled iterative estimates in the ResNet family of networks (Greff et al., 2017).

In the following, we will further investigate these candidates and their effects over performance collapse.

### A.2.1 HIGH LEVEL OF APPROXIMATION IN FIRST-ORDER DARTS

In (15), the optimality of $\omega^*(\alpha)$ w.r.t. the train loss is added to the formulation of the optimization problem of DARTS in the form of a constraint. Using the implicit function theorem, we can write the gradient corresponding to the outer optimization problem as (Zhang et al., 2021):

$$\nabla_\alpha \mathcal{L}^{val}(\omega^*(\alpha),\alpha) = \nabla_\alpha \mathcal{L}^{val}(\omega,\alpha) - \nabla^2_{\alpha,\omega} \mathcal{L}^{train}(\omega,\alpha) \left(\nabla^2_\omega \mathcal{L}^{train}(\omega,\alpha)\right)^{-1} \nabla_\omega \mathcal{L}^{val}(\omega,\alpha), \quad (19)$$

where $\omega = \omega^*(\alpha)$ and $\left(\nabla^2_\omega \mathcal{L}^{train}(\omega,\alpha)\right)^{-1}$ is the inverse of the Hessian w.r.t. $\omega$. In second-order DARTS, the inverse Hessian is estimated using the identity matrix, while in iDARTS the inverse Hessian is estimated using Neumann series. Note that it is entirely plausible for these approximations to be the cause of low layer alignment, and by extention, the performance collapse problem. So it may be the case that by eliminating at least one of these approximations in DARTS, the layer alignment may get improved.

In order to investigate this claim, we apply the layer alignment function $\Lambda(\omega,\alpha)$ over the layer gradients with the approximations removed:

$$\nabla_{\ell_p} \mathcal{L}^{val}(\omega^*(p),\alpha) = \nabla_{\ell_p} \mathcal{L}^{val}(\omega,\alpha) - \nabla^2_{\ell_p,\omega} \mathcal{L}^{train}(\omega,\alpha) \left(\nabla^2_\omega \mathcal{L}^{train}(\omega,\alpha)\right)^{-1} \nabla_\omega \mathcal{L}^{val}(\omega,\alpha), \quad (20)$$

where we can equivalently see $\omega^*(\alpha)$ as a function of $\omega^*(p)$ with the softmax function removed from the abstraction. Since we need to calculate (20) for each layer, we estimate the inverse Hessian with the identity function similar to second-order DARTS. Note that estimating (20) with this simplification can be performed similar to estimating the gradient corresponding $\Lambda(\omega,\alpha)$, by using finite difference approximation and perturbing the value of $p$ for each layer separately.

Comparing the average value for layer alignment for first-order and second-order estimation, we did not notice any meaningful differences. In both cases, the value of $\Lambda(\omega,\alpha)$ would vary between $0.06$ to $0.08$ on the NAS-Bench-201 search space, as can be observed in Figure-5, with the variation resembling the noise caused by mini-batching. Considering the performance of $\Lambda$-DARTS doesn't improve significantly until the layer alignment reaches a value larger than $0.2$ (as observed in Figure-1), we can confidently assert that the low layer alignment is not caused by the approximation of the gradient corresponding to architecture parameters.

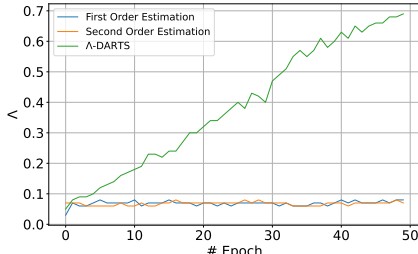

Figure 5: The average layer alignment for the first and second-order estimation of gradients of the layers, compared to $\Lambda$-DARTS.

### A.2.2 THE ITERATIVE ESTIMATION THEORY

An interesting observation to note is that the value of layer alignment is much lower for parametric operations compared to non-parametric operations. So it is completely plausible that the low layer alignment problem is somehow caused by the parametric operations (specifically, convolution operations) and their dynamic with the network in some way. Here, we will show that at according to the iterative estimation theory in ResNets (Greff et al., 2017), this is indeed the case.

According to (16), we can write the gradient corresponding to each operation on each edge for each layer as the dot product between the gradient of the node receiving the edge in that layer and the value of the operation itself. So we can write the correlation between the gradients of the layers $\ell$ and $\ell'$ for operation $o$ on edge $(i,j)$ as:

$$\frac{\partial \mathcal{L}}{\partial^\ell p_o^{(i,j)}} \cdot \frac{\partial \mathcal{L}}{\partial^{\ell'} p_o^{(i,j)}} = \frac{\partial \mathcal{L}}{\partial^\ell x_j}^T o(^\ell x_i) \cdot \frac{\partial \mathcal{L}}{\partial^{\ell'} x_j}^T o(^{\ell'} x_i) = o(^\ell x_i)^T \left[ \frac{\partial \mathcal{L}}{\partial^\ell x_j} \frac{\partial \mathcal{L}}{\partial^{\ell'} x_j}^T \right] o(^{\ell'} x_i), \quad (21)$$

which implicitly assumes the two layers work on the same number of height, width, and channels. Assuming $\frac{\partial \mathcal{L}}{\partial^\ell x_j}$ and $\frac{\partial \mathcal{L}}{\partial^{\ell'} x_j}$ are very similar (an assumption we will confirm empirically), then their outer product is very close to a positive semi-definite matrix. As a result, the correlation between layer gradients mainly depends on whether the value of the operations $o(^\ell x_i)$ and $o(^{\ell'} x_i)$ highly correlate or not.

Interestingly, the iterative estimation theory provides a framework for us to answer this question. According to this theory, the layers of a ResNet model try to iteratively refine an estimate of an ideal output that will ultimately result in an accurate prediction at the classification level (Greff et al., 2017; Jastrzebski

et al., 2018). As a result, we can see the output of each ResNet block as a representation in the same vector-space, with each block trying to move this representation in a direction that better estimates the ideal representation using its corresponding convolution operations.

Note that in this case, an optimal iterative estimation would correspond to a set of orthogonal updates, since undoing the work of previous blocks in undesirable. Furthermore, since most of the search spaces used for DARTS (and generally, NAS) are from the family of ResNets, we can apply these findings to DARTS as well. Therefore, under the iterative estimation theory, we can claim that the reason behind low layer alignment in convolution operations is the orthogonality of their output. This is inline with the motivation behind DARTS-PT in (Wang et al., 2021), which showed a direct connection between performance collapse and the iterative estimation theory. In this paper, we provide further evidence for this claim, but also provide a solution to the problem that does not require significant changes to the DARTS framework.

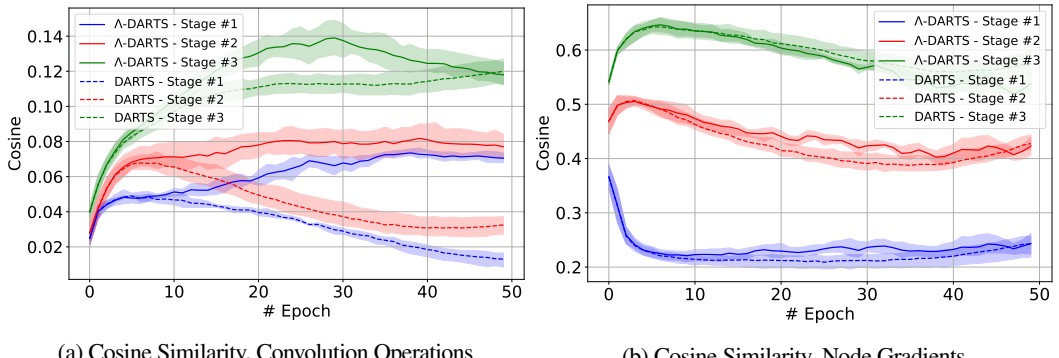

(a) Cosine Similarity, Convolution Operations          (b) Cosine Similarity, Node Gradients

Figure 6: The cosine similarity between (a) the convolution operations, (b) gradients received by each node, averaged over each pair of layers and each edge. We report the mean and the $95\%$ confidence interval for four experiments on the NAS-Bench-201 search space and the CIFAR-10 dataset.

In order to confirm our assumptions and provide empirical evidence for our analysis, here we will provide a visualization of the cosine similarity between $\frac{\partial \mathcal{L}}{\partial^{\ell} x_j}$ and $\frac{\partial \mathcal{L}}{\partial^{\ell'} x_j}$, and the convolution operation outputs $o(^{\ell} x_i)$ and $o(^{\ell'} x_i)$, averaged over each pair of layers on the NAS-Bench-201 search space. Since DARTS works in three stages with respect to input scale, we will provide the aforementioned values for the three stages separately. You can observe the results in Figure-6.

The first thing to note is the large correlation between node gradients, which increases in higher stages. On the otherhand, we can see that the correlation between the output of the convolution operations is incredibly low. This observation confirms our assumption that the main reason behind low layer alignment is low correlation between the output of the convolution operation, which can be attributed to the iterative estimation theory. Furthermore, we can observe that in order to increase the value of layer alignment, $\Lambda$-DARTS focuses on increasing the correlation between the output of convolutions, and does not have a noticeable impact on the correlation between node gradients.

Given the analytical and empirical evidence provided here, we can confidently assert that as pointed in (Wang et al., 2021), one of the main reasons behind performance collapse is the residual connections and the ResNet-like structure of most DARTS search spaces. But unlike (Wang et al., 2021), we attribute performance collapse to the discrepancies caused by the residual connections on the optimal architecture from the perspective of each layer, and the resulting problem in the convergence point of DARTS. The superior performance of $\Lambda$-DARTS compared to DARTS-PT confirms the validity of this claim, and we hope these findings can provide new directions for further research.

## A.3 RELATIONSHIP TO PRIOR WORKS

In this section, we will provide an analysis on the relationship between our work and some of the prior works that aim to solve the performance collapse problem. More specifically, we will show that most of the papers that reduce the severity of performance collapse to some degree can be explained through our analytical framework, suggesting that they may be alleviating the symptoms of low layer alignment indirectly.

Table 4: The test performance of architectures discovered by PC-DARTS with batch size of 256 (PC-DARTS-256) and 64 (PC-DARTS-64) on NAS-Bench-201.

| Method | CIFAR-10 Acc (%) | CIFAR-100 Acc (%) | ImageNet Acc (%) |
|---|---|---|---|
| PC-DARTS-256 | 93.41±0.30 | 67.48±0.89 | 41.30±0.22 |
| PC-DARTS-64 | 93.76±0.00 | 71.11±0.00 | 41.44±0.00 |

### A.3.1 THE CONVERGENCE OF DARTS AND ITS RELATION TO $\ell^2$ REGULARIZATION

In (Ye et al., 2022), it has been suggested that one of the possible solutions to performance collapse is $\ell^2$ regularization performed on the log-sum-exp of the architecture values: $\mathcal{L}_\beta(\alpha) = \log(\sum_{i<j, o\in\mathcal{O}} \exp(\alpha_o^{(i,j)}))$. The regularization term is then added to the loss function corresponding to the outer optimization problem, changing it to: $\min_\alpha \mathcal{L}(\omega^*(\alpha), \alpha) + \gamma \mathcal{L}_\beta(\alpha)$, where $\gamma$ is scheduled linearly, starting from 0 and increased to a very large value (usually around 25 to 100).

Note that in the case of $\ell^2$ regularization, the optimality condition in 1 can be re-written as:

$$\nabla_\alpha \mathcal{L}^{val}(\omega, \alpha^*) = -\gamma \nabla_\alpha \mathcal{L}_\beta(\alpha), \tag{22}$$

where we have $\nabla_\alpha \mathcal{L}_\beta(\alpha) = \sigma(\alpha) = p$. Given that $p$ is normalized over each edge, we have $\|\sigma(\alpha)\|_1 = |E|$. So we can given a lower-bound on the $\ell^2$ norm of $\nabla_\alpha \mathcal{L}_\beta(\alpha)$ using the Cauchy-Schwarz inequality:

$$\|\nabla_\alpha \mathcal{L}_\beta(\alpha)\|_2 \geq \frac{\|\nabla_\alpha \mathcal{L}_\beta(\alpha)\|_1}{\sqrt{|E|\cdot|\mathcal{O}|}} = \frac{|E|}{\sqrt{|E|\cdot|\mathcal{O}|}} = \sqrt{\frac{|E|}{|\mathcal{O}|}}. \tag{23}$$

Since the number of edges and operations are usually comparable in most search spaces (for example, in DARTS it is equal to 14 and 7, and in NAS-Bench-201 it is 6 and 5, respectively), the lower-bound proposed above is very close to 1. Therefore, as $\gamma$ increases to a very large value, the convergence point of $\beta$-DARTS no longer correspond to the stationary point of the primary loss function. Moreover, in our experiments we noticed that $\|\nabla_\alpha \mathcal{L}^{val}(\omega, \alpha)\|_2$ usually has a very small value (in the order of $O(10^{-1})$). This means that the convergence point doesn't happen on the saturation point of the softmax function, but instead, becomes dominated by the regularization term, which corresponds to the current value of $p$. So similar to the case of DARTS, the convergence point does not directly depend on the loss function, but is directed by the current value of $\alpha$ which is based on previous information from the gradient of the loss, which may be inaccurate at the current point.

This analysis is supported by the experiment results provided by (Ye et al., 2022) in its Figure-4c, where at the convergence point the value of $\alpha$ increases monotonically and linearly, while the value of $p$ does not change at all, and indeed does not correspond to the saturation point of softmax. Note that a similar analysis can be done on implicit gradient and second-order methods of DARTS as well, where the convergence point no longer corresponds to a stationary point of the explicit gradient of the architecture parameters

### A.3.2 VANISHING GRADIENT AND THE LOW LAYER ALIGNMENT ISSUE

In (Zhou et al., 2020) it has been observed that the rate of the convergence of the search model with respect to the operation parameters is directly impacted by the architecture weights corresponding to the skip-connection operations. This observation is then used an incentive to propose auxiliary skip-connections by (Chu et al., 2021) - dubbed DARTS-, where the effect of auxiliary connections is gradually reduced by a linearly scheduling their magnitude to approach zero. Independently, in order to reduce the memory consumption of the search model, PC-DARTS proposes performing the candidate operations on a subset of the channels of the input, thereby implicitly providing an auxiliary skip-connection on a subset of the input channels (Xu et al., 2020). Here, we will argue that both of these methods are implicitly trying to solve a symptom of the lower layer alignment issue by up-weighting the gradients corresponding to the lower layers.

Firstly, in order to show that the increased batch size doesn't benefit PC-DARTS greatly, we performed the search using PC-DARTS on a batch size of 64 and compared the results to performing the search with a batch size of 256 on the NAS-Bench-201 search space in Table-4. Clearly, we can claim that the batch size is not the main contributer to the better performance of PC-DARTS, and instead, it is the partial-channeling that appears to be alleviating performance collapse.

Note that according to (16), we can write the gradient received by each architecture weight as the dot product of the gradient of the receiving node and the corresponding operation. While gradient vanishing

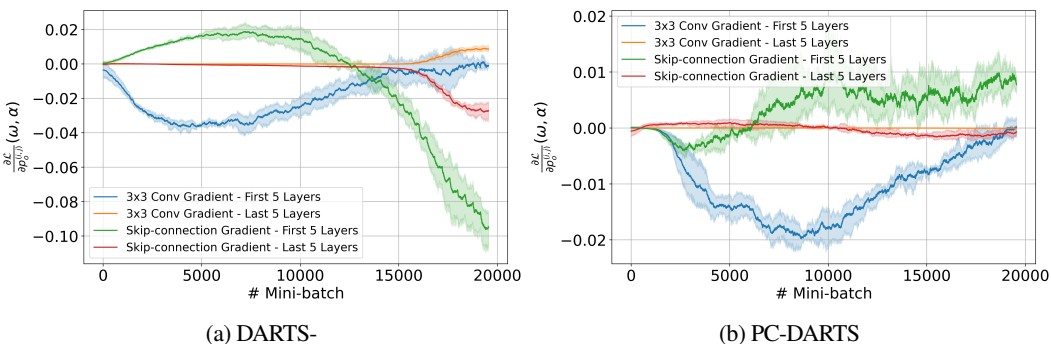

|                |                |
|:--------------:|:--------------:|
| (a) DARTS-     | (b) PC-DARTS   |

Figure 7: The exponential average of the gradient of the first five layers and the last five layers for DARTS- and PC-DARTS, similar to Figure-2.

does not affect the operation, it does have an impact over the gradient of the loss w.r.t. the receiving node. We can see this effect in Figure-2, wherein the gradients corresponding to the first 5 layers generally have smaller magnitudes. Therefore, one way of alleviating the symptoms of low layer alignment can be reducing the effects of gradient vanishing by providing stronger skip-connections. This is effectively what is done by the auxiliary skip-connection proposed by DARTS- and the partial-channel structure of PC-DARTS, which causes limited improvements in the performance of DARTS.

In order to make our argument stronger, we also provide the gradients corresponding to the first five layers and the last five layers similar to Figure-2 for both DARTS- and PC-DARTS, which you can see in Figure-7. The first thing to notice is the striking similarity between the two figures, where the gradients corresponding to layers closer to the output have been extremely down-weighted compared to the gradients corresponding to layers closer to the input. This observation proves the validity of our claim that auxiliary skip-connection and partial-channeling significantly up-weights the gradients corresponding to the layers closer to the input. Since these layers prefer parametric operations, we observe better performance by these models. But note that despite the improvements, these methods are merely circumventing the problem of low layer alignment, which is the reason behind their poor performance compared to $\Lambda$-DARTS.

### A.3.3 HIGH LAYER ALIGNMENT AS A PRIOR BY ELIMINATING WEIGHT-SHARING IN SAMPLE-BASED NAS

One of the popular methods of cell-based NAS is the use of sampling candidate architectures from a pre-determined distribution, wherein the sufficient statistics of the distribution are optimized according to the performance of the sampled architecture. Among these methods, we can mention DrNAS which utilizes a Dirichlet distribution for the search (Chen et al., 2021a), and SNAS which utilizes a concrete distribution (Xie et al., 2019). In both of these cases, the sufficient statistics are optimized according to a gradient estimation technique that approximates the gradient corresponding to the sufficient statistics based on the gradients received by the sampled operations.

We argue that the mere act of using sample-based NAS is a way to circumvent the problem of low layer alignment. Firstly, because such a formulation of the NAS problem completely eliminates the issue of convergence at softmax saturation points, thus preventing the optimal architecture to be completely dictated by the layers closer to the input. Secondly, because the sampled architecture in a cell-based NAS method is used on all of the cells, sampling itself comes with an implicit prior that the optimal architecture according to each layer must be the same. Therefore, sample-based NAS can be also viewed as a way to enforce high layer alignment in an a priori fashion. Therefore, we can see sampled-based differentiable NAS as a way of solving the problem of performance collapse. But we should note that since in these methods only a subset of the supernet is optimized, they usually come with the caveat of low sample efficiency and longer search time (Ren et al., 2021).

### A.4 REGULARIZATION OF THE OUTER OBJECTIVE

An interesting point to consider is whether we can apply the regularization term of $\Lambda$-DARTS over the outer objective, which will directly optimize the architecture parameters $\alpha$. Here, we will first investigate

Table 5: The test performance of architectures discovered by $\Lambda$-DARTS with the regularization term applied to the inner objective (inner), and the regularization term applied to the outer objective (outer). The search is performed on CIFAR-10 and the NAS-Bench-201, and the mean and standard deviation of three runs is reported.

| Model | CIFAR-10 Acc (%) | CIFAR-100 Acc (%) | ImageNet Acc (%) |
|---|---|---|---|
| $\Lambda$-DARTS (inner) | 93.95$\pm$0.15 | 72.75$\pm$0.42 | 45.49$\pm$0.08 |
| $\Lambda$-DARTS (outer) | 91.37$\pm$1.59 | 67.87$\pm$1.57 | 41.6$\pm$2.11 |

Table 6: The test performance of P-DARTS, P-DARTS- (P-DARTS with auxiliary connections), and P-DARTS with the proposed regularization term. In all experiments, the prior assumptions of P-DARTS are removed. The search is performed on CIFAR-10 and the DARTS search space, and the mean and standard deviation of three runs is reported.

| Model | CIFAR-10 Acc (%) |
|---|---|
| P-DARTS (w/o priors) (Chen et al., 2021b) | 96.48$\pm$0.55 |
| P-DARTS- (w/o priors) (Chen et al., 2021b) | 97.28$\pm$0.04 |
| $\Lambda$-P-DARTS (w/o prios) | **97.34$\pm$0.06** |

this matter from a theoretical standpoint, and provide two points that shows the sub-optimality of such approach. Then provide empirical results to confirm our analytical findings.

The first problem with regularizing the outer objective has to do with the fact that the gradient of $p_o^{(i,j)}$ does not directly depend on $p_o^{(i,j)}$ itself, but the output of the corresponding operation $o(.)$ and the gradient of the receiving node $x_j$, as observed in (16). Therefore, increasing the layer alignment must be done by changing the composition of the operations of the layers in the higher layers to adjust $\frac{\partial \mathcal{L}}{\partial^\ell x_j}$, while changing the composition of the operations in the lower layers to adjust $o(^\ell x_i)$. Considering the small size of $\alpha$, and the weight-sharing framework, this may be an impossible task that may result in overfitting on the regularization term as well.

Another issue we need to take note of is the connection between the problem of low layer alignment and the iterative estimation theory, as noted in Appendix-A.2. Assuming the problem of low layer alignment is caused by the orthogonality of the output of convolution operations in each layer, then resolving this problem is only possible through regularizing the convolution operations themselves. While it may be possible to resolve the issue through adjusting the input, we note that as pointed in the previous paragraph, this adjustment may lead to overfitting due to the low complexity of the model with respect to $\alpha$.

In order to provide empirical evidence for our analysis, we tried to perform the regularization on the outer objective. You can see the results of 50 epochs of search in Table-5, where we compared $\Lambda$-DARTS with inner objective regularized with $\Lambda$-DARTS with outer objective regularized. Clearly, we observe that regularizing the outer objective performs poorly compared to the case where we regularized the inner objective. This observation further supports our analytical reasoning, and shows that performing the regularization on $\omega$ yields more superior and robust performances compared to regularizing $\alpha$.

## A.5 COMBINATION WITH ORTHOGONAL VARIANTS OF DARTS

In this section, we will empirically demonstrate that our method can be extended to orthogonal variants of DARTS to yield better and more robust results. Specifically, we will discuss two popular variants of DARTS, P-DARTS and SDARTS, and how $\Lambda$-DARTS can eliminate the need for strong prior assumptions or costly regularization in these methods.

### A.5.1 PROGRESSIVE-DARTS AND THE PROBLEM OF STRONG PRIOR ASSUMPTIONS

Progressive-DARTS (P-DARTS) is a low-cost variant of DARTS that is based on performing the search in a progressive manner. P-DARTS performs the search in three stages, with each stage eliminating some of the candidate operations while also making the search model deeper and more similar to the evaluation architecture (Chen et al., 2021b). One of the issues with PC-DARTS is its reliance on two very strong priors in the search process: 1) limiting the number of skip-connections to two, which severely reduces the size of

Table 7: The test performance of SDARTS-RS, SDARTS-ADV, and SDARTS-RS with the proposed regularization term. The search and evaluation is performed on CIFAR-10 and the DARTS search space, and the mean and standard deviation of three runs is reported. The search cost is reported based on GPU days on a 1080 Ti.

| Model | Test Acc (%) | Search Cost (GPU days) |
|---|---|---|
| SDARTS-RS (Chen & Hsieh, 2020) | 97.33±0.03 | 0.4 |
| SDARTS-ADV (Chen et al., 2021b) | 97.39±0.02 | 1.3 |
| $\Lambda$-SDARTS-RS | **97.42±0.04** | 0.8 |

the search space, 2) using dropout on skip-connections to limit their utility during search. Following (Chu et al., 2021), here we will show that $\Lambda$-DARTS can eliminate the need for both of these strong priors, which further validates the effectiveness of our method. You can see the results in Table-6.

We can see that our method successfully removes any need for strong prior assumptions or auxiliary connections in the progressive search format, achieving superior performance to both cases. We note that these results are attained without any adjustment to the hyperparameters of $\Lambda$-DARTS, which is likely going to further benefit the results, considering that the progressive framework changes the macro-architecture and the search space significantly in each stage.

### A.5.2 SDARTS AND THE PROBLEM OF COSTLY REGULARIZATION

SDARTS is a variant of DARTS that aims to improve generalization by regularizing the Hessian of the loss function corresponding to the architecture parameters (Chen & Hsieh, 2020). The regularization is done through additive noise to the architecture parameters during the optimization of operation parameters ($\omega$). There are tow methods proposed by (Chen & Hsieh, 2020) based on this idea: 1) SDARTS-RS and 2) SDARTS-ADV. SDARTS-RS is based on random perturbations sampled from a uniform distribution, while SDARTS-ADV utilizes the costly method of projected gradient descent to produce adversarial perturbations. One of the issues with SDARTS-RS is that it is not as effective as the costly method of SDARTS-ADV, which needs more than four forward-backward propagations to estimate an effective perturbation. Here, we will show that by combining our proposed method with SDARTS-RS, we will achieve better results than both SDARTS-RS and SDARTS-ADV, at the cost of only two forward-backward propagations.

In Table-7, you can see the performance of SDARTS with and without adversarial perturbations, and compare them to that of $\Lambda$-SDARTS-RS, which utilizes our proposed regularization term along with the random perturbation regularization of SDARTS-RS. Clearly, our proposed regularization completely eliminates the need for costly adversarial training, reducing the cost to almost half of what SDARTS-ADV requires. This observation further proves the importance of increasing the layer alignment in DARTS, regardless of the type of DARTS variant used during search. We note that these results are attained without any adjustment to the hyperparameters of $\Lambda$-DARTS, which is likely going to further benefit the results, considering that the Hessian regularization is likely to change the optimal $\epsilon_0$ due to changes in the loss landscape.

### A.6 PERFORMANCE COLLAPSE WITH MORE EPOCHS

Recently, it has been shown that performance collapse happens more consistently and frequently the longer the search process is performed (Bi et al., 2019). This observation is inline with our analysis, which shows the negative impacts of low layer alignment becomes more pronounced with longer epochs of search, which is attributed to the softmax saturation point and vanishing gradients. In order to show that $\Lambda$-DARTS has successfully alleviated this problem even in the most complex search spaces, we performed the search on the DARTS search space for 200 epochs following (Bi et al., 2019). You can see the results in Table-8.

Clearly, our method successfully mitigates performance collapse even in the most complex search spaces. Among the baselines, the only method with comparable performance to $\Lambda$-DARTS is Amended-DARTS, which performs the search at $50\%$ more search cost due to its second-order nature. Furthermore, comparing the number of parametric operations, we can see that our method is much better at discovering efficient architecture that are not unnecessarily over-parameterized. On the otherhand, Amended-DARTS perform poorly despite the fact that it has two more parametric operations. Furthermore, we can see that the

Table 8: The test performance of Random Search, DARTS (first and second order), P-DARTS, PC-DARTS, Amended-DARTS, and Λ-DARTS after 200 epochs of search on CIFAR-10. All results, except for Λ-DARTS, are borrowed from (Chen et al., 2021b). The search cost is reported based on GPU days on a 1080 Ti, and inferred from reported costs for the baselines. Note that here, (#P) corresponds to the number of parametric operations in the normal cell.

| Model | Test Acc (%) (%) | #P | Search Cost (GPU days) |
|---|---|---|---|
| Random Search (Bi et al., 2019) | 96.71 | - | - |
| DARTS (1st) (Liu et al., 2019) | 93.82 | 0 | 1.6 |
| DARTS (2nd) (Liu et al., 2019) | 94.85 | 0 | 4.0 |
| P-DARTS (Chen et al., 2021b) | 94.62 | 0 | 1.2 |
| PC-DARTS (Xu et al., 2020) | 96.85 | 3 | 0.4 |
| Amended-DARTS (Bi et al., 2019) | 97.29 | 7 | 4.8 |
| Λ-DARTS | **97.34** | 5 | 3.2 |

Table 9: The test performance of Λ-DARTS, compared with baselines, with the search and evaluation performed on ImageNet and the DARTS search space.

| Model | ImageNet Top-1 Acc (%) | ImageNet Top-5 Acc (%) |
|---|---|---|
| NASNet-A (Zoph et al., 2018) | 74.0 | 91.6 |
| DARTS (2nd) (Liu et al., 2019) | 73.3 | 91.3 |
| SNAS (Xie et al., 2019) | 72.7 | 90.8 |
| GDAS (Dong & Yang, 2019) | 74.0 | 91.5 |
| P-DARTS (Chen et al., 2021b) | 75.3 | 92.5 |
| PC-DARTS (Xu et al., 2020) | 75.8 | 92.7 |
| DrNAS (Chen et al., 2021a) | 75.8 | 92.7 |
| SDARTS-ADV (Chen & Hsieh, 2020) | 74.8 | 92.2 |
| DOTS (Gu et al., 2021) | 75.7 | 92.6 |
| DARTS+PT (Wang et al., 2021) | 74.5 | 92.0 |
| $\beta$-DARTS (Ye et al., 2022) | 75.8 | 92.9 |
| Λ-DARTS-Λ$(\omega,\alpha)$ | **75.9** | **93.0** |

only models performing better than random baseline are PC-DARTS, Amended-DARTS, and Λ-DARTS. Considering the connection between Λ-DARTS and PC-DARTS, as pointed out at Appendix-A.3, this observation further proves the role of vanishing gradient and low layer alignment in performance collapse.

## A.7 SEARCH RESULTS ON IMAGENET

In this section, we provide the search and evaluation results for the ImageNet dataset. The search was performed on the setting proposed by PC-DARTS (Xu et al., 2020), on a search model with 8 layers (6 normal and 2 reduction cells) and 16 initial channels. Following PC-DARTS we randomly sampled $10\%$ and $2.5\%$ of the ImageNet dataset for optimizing $\omega$ and $\alpha$, respectively. We used a batch size of $384$ and a learning rate of $0.5$ which is linearly decayed to $0.0$. The evaluation was performed with a similar setting, for $250$ epochs with a model with $14$ layers and $48$ initial channels. You can see the results in Table-9.

We can see that our proposed method improves upon all baselines both in terms of top-1 and top-5 accuracy. We improve upon the performance of DARTS by $2.5\%$ in terms of top-1 accuracy and $1.7\%$ in terms of top-5 accuracy. Furthermore, we beat the best-performing baseline, namely $\beta$-DARTS, by a margin of $0.1\%$ in terms of top-1 accuracy and $0.1\%$ in terms of top-5 accuracy. We also note that DrNAS utilizes a slightly different setting for search, which performs the search on a search model with 14 layers and 48 initial channels, similar to the evaluation model. Despite this advantage, our proposed method beats the performance of DrNAS by a margin of $0.1\%$ on top-1 accuracy and $0.3\%$ on top-5 accuracy. The discovered architectures can be seen in Figure-22.

Table 10: The test performance of architectures discovered by Λ-DARTS with central finite difference approximation (ctr) and forward/backward finite difference approximation (f/b) for 50 epochs of search on CIFAR-10.

| Method | CIFAR-10 Acc (%) | CIFAR-100 Acc (%) | ImageNet Acc (%) |
|---|---|---|---|
| Λ-DARTS (ctr) | 94.04±0.03 | 72.75±0.42 | 45.49±0.08 |
| Λ-DARTS (f/b) | 93.95±0.15 | 72.33±0.71 | 44.98±0.90 |

## A.8 SEARCH COST

One of the important factors in NAS is the cost of performing the search on a large-scale dataset, which is usually measured in terms of GPU hours or GPU days on the DARTS search space and the CIFAR-10 dataset (Elsken et al., 2019). Since Λ-DARTS requires two extra forward-backward passes to estimate the gradients of the regularization term, it costs twice as much as DARTS to perform the search on the same dataset. Therefore, the cost of our method in terms of GPU days is $0.8$ days on a GTX 1080 Ti GPU on the DARTS search space and CIFAR-10 dataset, which is $20\%$ less than the second-order DARTS (at $1.0$ GPU days) and about $40\%$ less than SDARTS-ADV (at about $1.3$ GPU days), both of which perform inferior to our model.

In order to reduce the search cost and make our method comparable to most baselines, we can perform forward or backward finite difference approximation instead of a central difference approximation (Shen et al., 2020):

$$\nabla_\omega^{fwd}\Lambda(\omega,\alpha) = \frac{1}{\binom{L}{2}}\left[\frac{\nabla_\omega\mathcal{L}(\omega,\boldsymbol{P}+\epsilon\Delta)-\nabla_\omega\mathcal{L}(\omega,\boldsymbol{P})}{\epsilon}\right]+O(\epsilon),\tag{24}$$

$$\nabla_\omega^{bwd}\Lambda(\omega,\alpha) = \frac{1}{\binom{L}{2}}\left[\frac{\nabla_\omega\mathcal{L}(\omega,\boldsymbol{P})-\nabla_\omega\mathcal{L}(\omega,\boldsymbol{P}-\epsilon\Delta)}{\epsilon}\right]+O(\epsilon).\tag{25}$$

Note that since $\nabla_\omega^{fwd}\Lambda(\omega,\alpha)$ and $\nabla_\omega^{bwd}\Lambda(\omega,\alpha)$ are biased estimations, we can use them in an interleaved fashion, where we perform forward estimation for one mini-batch, and a backward estimation in the next. This will reduce the cost by $25\%$, putting our method at around $0.6$ GPU days, which is only $50\%$ more than the original first-order DARTS (at $0.4$ GPU days), and comparable to DrNAS with progressive search. You can see a thorough comparison with state-of-the-art NAS models in terms of performance on CIFAR-10 and search cost in Table-11.

In order to show the reduction in the accuracy of the estimation does not reduce the effectiveness of Λ-DARTS, we have provided the results of discovered architectures by this low-cost method compared to the central method for 50 epochs on the NAS-Bench-201 search space on CIFAR-10 in Table-10. We can see that in all three datasets, the drop in average performance is negligible (less than $0.1\%$ in CIFAR-10 and less than $0.5\%$ in CIFAR-100 and ImageNet). Therefore, it is clear that using forward/backward finite difference approximation can be a good technique to reduce the cost of searching on large datasets, which makes our method comparable to DrNAS in terms of cost for large scale applications.

## A.9 EXPERIMENT SETTINGS

### A.9.1 NAS-BENCH-201 SEARCH SPACE

NAS-Bench-201 provides a DARTS-like cell-search framework, where we search for a single cell containing 3 nodes and 5 operations. The performance of the discovered architecture on CIFAR-10, CIFAR-100, and ImageNet16-120 datasets can be attained by querrying the database provided by (Dong & Yang, 2020). We used the implementation of (Dong & Yang, 2020) for our search. The operation parameters ($\omega$) are optimized using stochastic gradient descent with Nestrov momentum. The learning rate is set to $0.025$, gradually reduced to $0.001$ using cosine scheduling. The weight decay and momentum are set to $0.0005$ and $0.9$, respectively.

For architecture parameters ($\alpha$), we use Adam with the learning rate set to $10^{-4}$ and the weight decay rate set to $0.001$. The momentum values $\beta_1$ and $\beta_2$ are set to $0.5$ and $0.999$, respectively. We perform the search on CIFAR-10 for 100 epochs, and we set $\epsilon_0 = 0.0001$ and $\lambda = 0.125$ for this experiment.

Table 11: Comparison with state-of-the-art NAS models on the CIFAR-10 dataset. We provide the search cost based on GPU days on a 1080 Ti.

| Method | Test Acc (%) | Params (M) | Search Cost (GPU days) |
|---|---|---|---|
| NASNet-A (Zoph et al., 2018) | 97.35 | 3.3 | 2000 |
| ENAS (Pham et al., 2018) | 97.11 | 4.6 | 0.5 |
| DARTS (1st) (Liu et al., 2019) | 97.00±0.14 | 3.4 | 0.4 |
| DARTS (2nd) (Liu et al., 2019) | 97.24±0.09 | 3.3 | 1.0 |
| SNAS (Xie et al., 2019) | 97.15±0.02 | 2.8 | 1.5 |
| GDAS (Dong & Yang, 2019) | 97.07 | 3.4 | 0.3 |
| P-DARTS (Liu et al., 2018) | 97.50 | 3.4 | 0.3 |
| R-DARTS (Zela et al., 2020) | 97.05±0.21 | - | 1.6 |
| PC-DARTS (Xu et al., 2020) | 97.43±0.07 | 3.6 | 0.1 |
| DrNAS (Chen et al., 2021a) | 97.46±0.03 | 4.0 | 0.4 |
| SDARTS-ADV (Chen & Hsieh, 2020) | 97.39±0.02 | 3.3 | 1.3 |
| DARTS- (Chu et al., 2021) | 97.41±0.08 | 3.5 | 0.4 |
| DARTS+PT (Wang et al., 2021) | 97.39±0.08 | 3.0 | 0.8 |
| $\Lambda$-DARTS - $\Lambda(\omega,\alpha)$ | 97.48±0.11 | 3.5 | 0.8 |
| $\Lambda$-DARTS - $\Lambda_{\pm}(\omega,\alpha)$ | **97.57±0.05** | 3.6 | 0.8 |

### A.9.2 DARTS SEARCH SPACE

The search space of DARTS contains two types of cells: Normal and Reduction, corresponding to layers with stride of 1 and 2, respectively. Each cell has 4 intermediate nodes and 7 operations. Here, we only apply the regularization term to the normal cell, since the performance collapse is not usually associated with the reduction cell (Zela et al., 2020). We used the implementation provided by (Wang et al., 2021; Chen & Hsieh, 2020). The operation parameters ($\omega$) are optimized using stochastic gradient descent with momentum. The learning rate is set to 0.025, gradually reduced to 0.001 using cosine scheduling. The weight decay and momentum are set to 0.0003 and 0.9, respectively.

For architecture parameters ($\alpha$), we use Adam with the learning rate set to $3 \times 10^{-4}$ and the weight decay rate set to 0.001. The momentum values $\beta_1$ and $\beta_2$ are set to 0.5 and 0.999, respectively. We perform the search on CIFAR-10 for 50 epochs, and we set the value for $\lambda$ to 0.25 for $\Lambda(\omega,\alpha)$ and 0.125 for $\Lambda_{\pm}(\omega,\alpha)$. In both cases, we set the value of $\epsilon_0$ to 0.0001.

### A.9.3 REDUCED SEARCH SPACES

The four search spaces proposed by (Zela et al., 2020) contain some subset of the operations used in the DARTS search space, except for the fourth search space which contains convolutions and random noise as operations. Otherwise, the cells have the same structure as in the DARTS search space.

We perform the search and evaluation on three different datasets: CIFAR-10, CIFAR-100, and SVHN. For CIFAR-10, the search setting is the same as in Section-5.2. For SVHN and CIFAR-100, we increase the batch size and the value of $\epsilon_0$ to 96 and 0.001, respectively. The evaluation setting for these datasets is the same as in (Zela et al., 2020). For all experiments, we only report the results of $\Lambda(\omega,\alpha)$, since both methods perform similarly.

### A.9.4 ABLATION STUDY

For this experiment, we used the exact same setting as in Section-5.1, except for the architecture learning rate. For the experiments on $\epsilon_0$ and $\lambda$, the architecture learning rate is set to $3 \times 10^{-4}$. For the experiments on the number of epochs, the architecture learning rate is set to $3 \times 10^{-4}$ when performing the search for 50 epochs, and reduced to $10^{-4}$ when performing the search for more than 50 epochs.

For experiments corresponding to $\epsilon_0$, we set $\lambda = 0.125$, while for experiments corresponding to $\lambda$, we set $\epsilon_0 = 0.0001$, and for experiments corresponding to the number of epochs, we set $\epsilon_0 = 0.0001$ and $\lambda = 0.125$.

## A.10 Proof of Proposition 1

**Proof:** Using simple linear algebra, we can re-write the squared norm of the gradient $\nabla_\alpha \mathcal{L}(\omega,\alpha)$ as:

$$\|\nabla_\alpha \mathcal{L}(\omega,\alpha)\|_2^2 = \|\boldsymbol{J}_\sigma(\alpha)\nabla_{\boldsymbol{p}}\mathcal{L}(\omega,\alpha)\|_2^2 = \nabla_{\boldsymbol{p}}\mathcal{L}(\omega,\alpha)^T \boldsymbol{J}_\sigma(\alpha)^T \boldsymbol{J}_\sigma(\alpha)\nabla_{\boldsymbol{p}}\mathcal{L}(\omega,\alpha). \tag{26}$$

Now using the assumption that $\nabla_{\boldsymbol{p}}\mathcal{L}(\omega,\alpha)$ is the sum of vectors orthogonal to the null-space of the Jacobian matrix and the fact that $\boldsymbol{J}_\sigma(\alpha)^T \boldsymbol{J}_\sigma(\alpha)$ is a positive semi-definite matrix, we can give the following lower-bound for this term (Meyer, 2000):

$$\nabla_{\boldsymbol{p}}\mathcal{L}(\omega,\alpha)^T \boldsymbol{J}_\sigma(\alpha)^T \boldsymbol{J}_\sigma(\alpha)\nabla_{\boldsymbol{p}}\mathcal{L}(\omega,\alpha) \geq \min_{i,\lambda_i(\alpha)\neq 0} \lambda_i^2 \cdot \|\nabla_{\boldsymbol{p}}\mathcal{L}(\omega,\alpha)\|_2^2. \tag{27}$$

Since we assume the gradient $\nabla_{\boldsymbol{p}}\mathcal{L}(\omega,\alpha)$ can be written as the sum of orthogonal layer gradients, its norm can be lower bounded by the smallest gradient (Meyer, 2000):

$$\|\nabla_{\boldsymbol{p}}\mathcal{L}(\omega,\alpha)\|_2^2 = \|\sum_{\ell=1}^{L}\nabla_{\ell\boldsymbol{p}}\mathcal{L}(\omega,\alpha)\|_2^2 \tag{28}$$

$$= \sum_{\ell=1}^{L}\|\nabla_{\ell\boldsymbol{p}}\mathcal{L}(\omega,\alpha)\|_2^2 \tag{29}$$

$$\geq L\cdot\min_{\ell}\|\nabla_{\ell\boldsymbol{p}}\mathcal{L}(\omega,\alpha)\|_2^2, \tag{30}$$

where 29 follows from the orthogonality. So using (27) and (30), we can get:

$$\|\nabla_\alpha \mathcal{L}(\omega,\alpha)\|_2^2 \geq \min_{i,\lambda_i(\alpha)\neq 0} \lambda_i^2 \cdot L\cdot\min_{\ell}\|\nabla_{\ell\boldsymbol{p}}\mathcal{L}(\omega,\alpha)\|_2^2. \tag{31}$$

## A.11 Discovered architectures

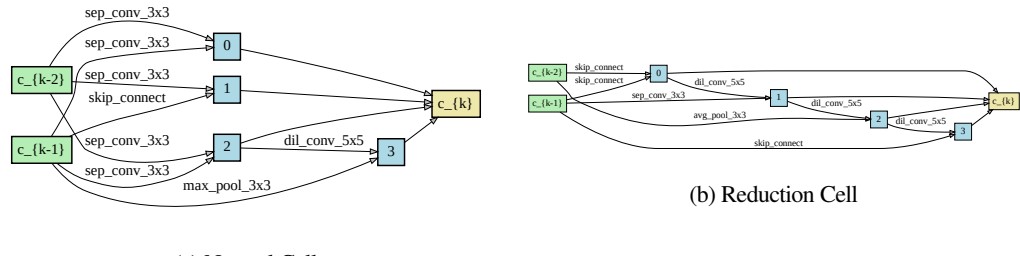

(a) Normal Cell

(b) Reduction Cell

Figure 8: Best normal and reduction cells discovered by $\Lambda(\omega,\alpha)$ on DARTS search space and CIFAR-10 dataset.

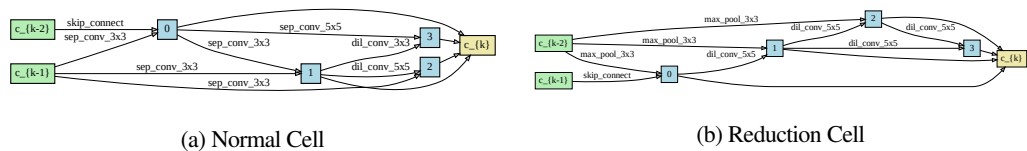

(a) Normal Cell

(b) Reduction Cell

Figure 9: Best normal and reduction cells discovered by $\Lambda_\pm(\omega,\alpha)$ on DARTS search space and CIFAR-10 dataset.

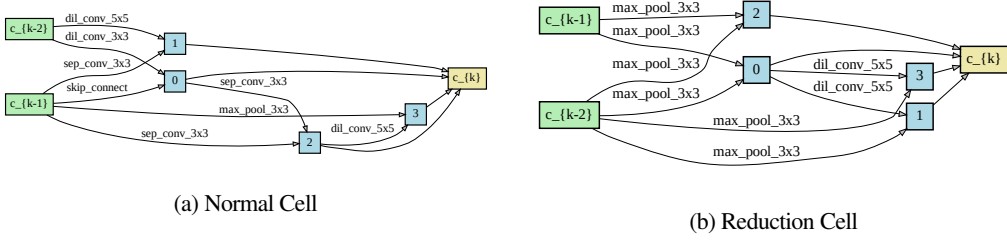

(a) Normal Cell

(b) Reduction Cell

Figure 10: Best normal and reduction cells discovered by $\Lambda(\omega,\alpha)$ on S1 and CIFAR-10 dataset.

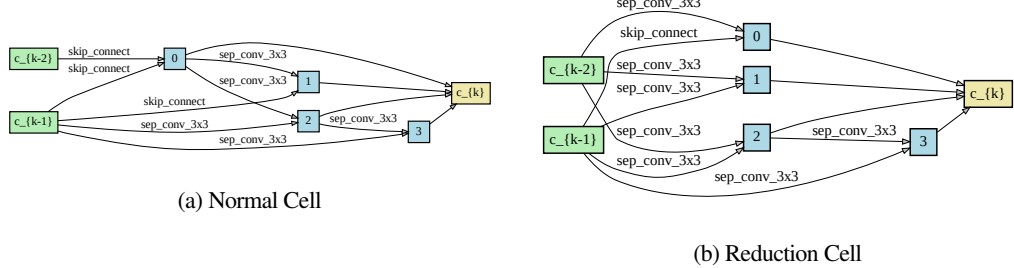

(a) Normal Cell

(b) Reduction Cell

Figure 11: Best normal and reduction cells discovered by $\Lambda(\omega,\alpha)$ on S2 and CIFAR-10 dataset.

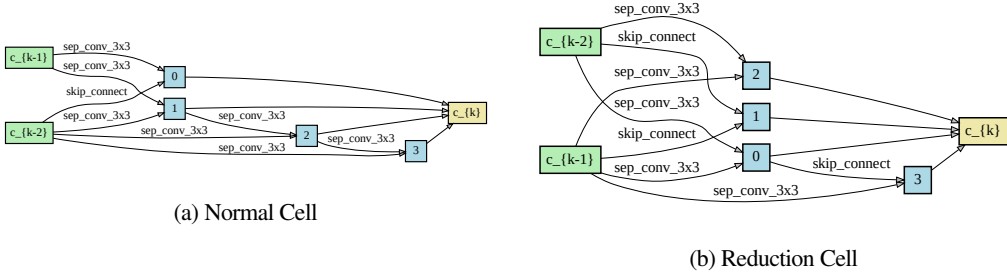

(a) Normal Cell

(b) Reduction Cell

Figure 12: Best normal and reduction cells discovered by $\Lambda(\omega,\alpha)$ on S3 and CIFAR-10 dataset.

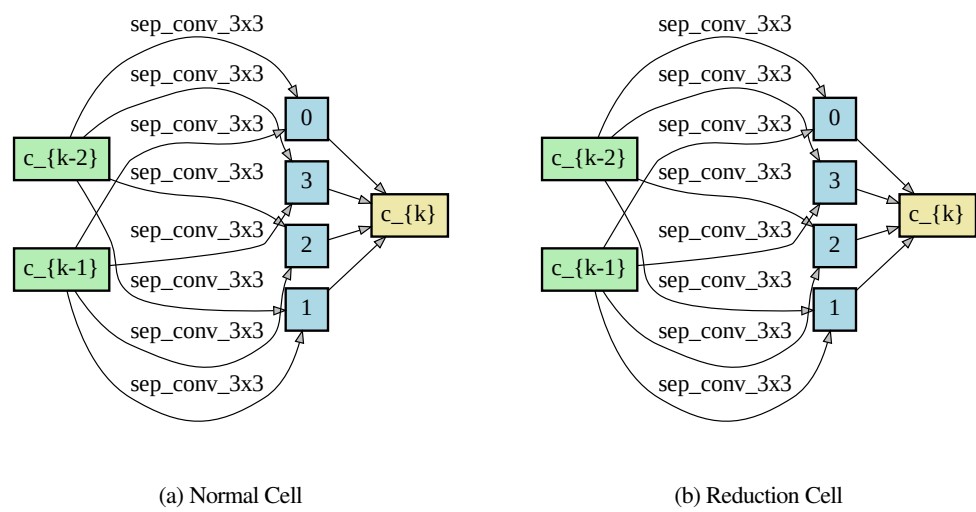

(a) Normal Cell

(b) Reduction Cell

Figure 13: Best normal and reduction cells discovered by $\Lambda(\omega,\alpha)$ on S4 and CIFAR-10 dataset.

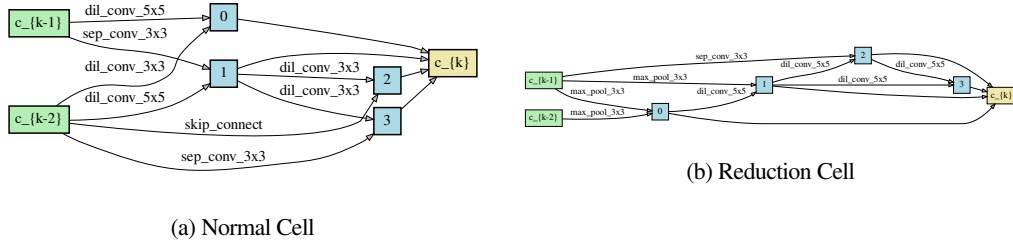

(a) Normal Cell

(b) Reduction Cell

Figure 14: Best normal and reduction cells discovered by $\Lambda(\omega,\alpha)$ on S1 and CIFAR-100 dataset.

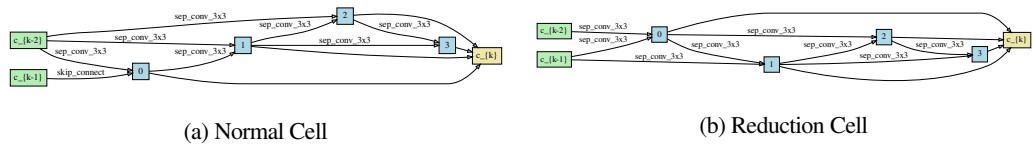

(a) Normal Cell

(b) Reduction Cell

Figure 15: Best normal and reduction cells discovered by $\Lambda(\omega,\alpha)$ on S2 and CIFAR-100 dataset.

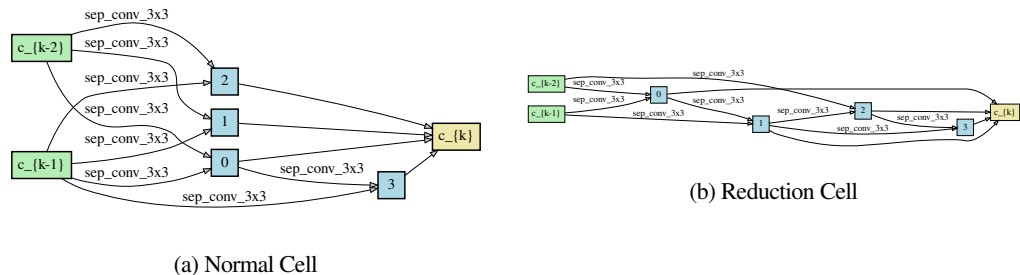

(a) Normal Cell

(b) Reduction Cell

Figure 16: Best normal and reduction cells discovered by $\Lambda(\omega,\alpha)$ on S3 and CIFAR-100 dataset.

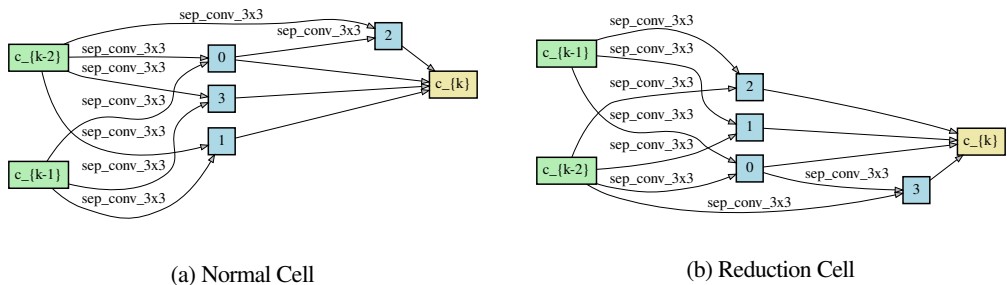

(a) Normal Cell

(b) Reduction Cell

Figure 17: Best normal and reduction cells discovered by $\Lambda(\omega,\alpha)$ on S4 and CIFAR-100 dataset.

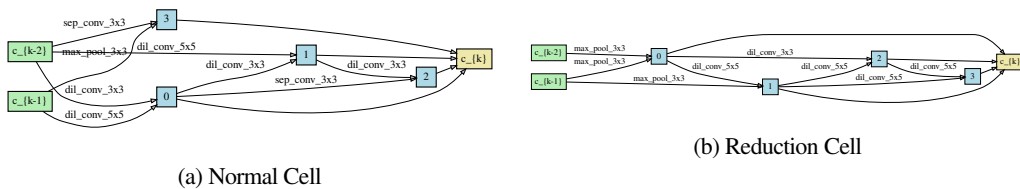

(a) Normal Cell

(b) Reduction Cell

Figure 18: Best normal and reduction cells discovered by $\Lambda(\omega,\alpha)$ on S1 and SVHN dataset.

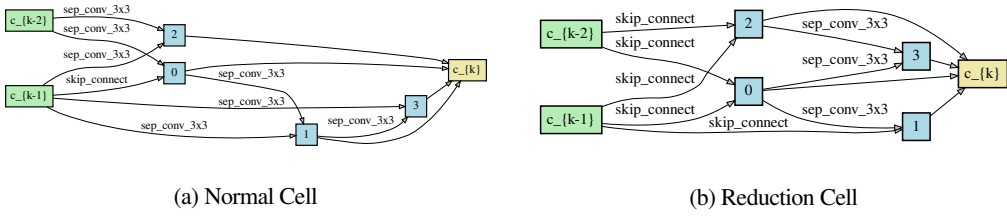

(a) Normal Cell        (b) Reduction Cell

Figure 19: Best normal and reduction cells discovered by $\Lambda(\omega,\alpha)$ on S2 and SVHN dataset.

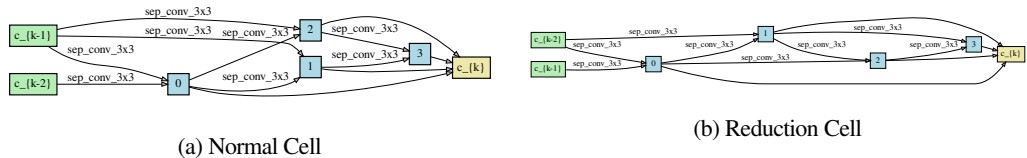

(a) Normal Cell        (b) Reduction Cell

Figure 20: Best normal and reduction cells discovered by $\Lambda(\omega,\alpha)$ on S3 and SVHN dataset.

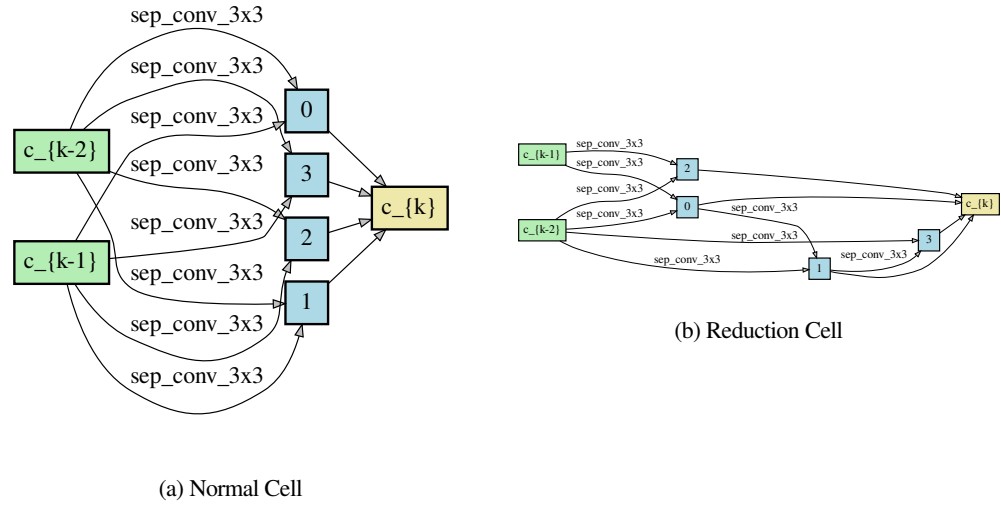

(a) Normal Cell

Figure 21: Best normal and reduction cells discovered by $\Lambda(\omega,\alpha)$ on S4 and SVHN dataset.

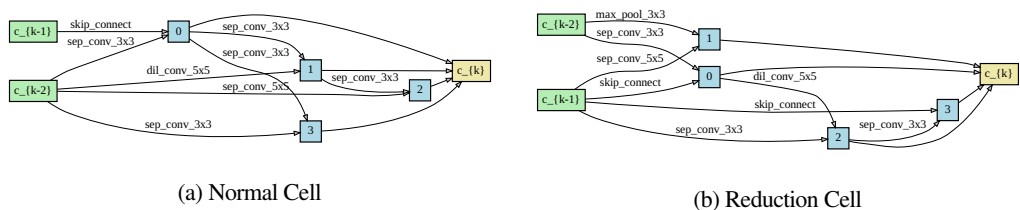

(a) Normal Cell        (b) Reduction Cell

Figure 22: Best normal and reduction cells discovered by $\Lambda(\omega,\alpha)$ on DARTS search space and the ImageNet dataset.

