# OpenReview forum: "$\Lambda$-DARTS: Mitigating Performance Collapse by Harmonizing Operation Selection among Cells"
_ICLR.cc/2023/Conference — ICLR 2023 poster_

### Official Review · Reviewer_iL42 · 2022-10-23

**Confidence:** 5
**Correctness:** 2
**Technical Novelty And Significance:** 2
**Empirical Novelty And Significance:** 3
**Recommendation:** 6

**Clarity, Quality, Novelty And Reproducibility:**

Clarity: good.

Quality: the weakness mainly lies in the justification of the proposed explanation.

Novelty: moderate -- the proposed explanation for collapse looks new, but it was not well validated.

Reproducibility: code was not provided, and the formulation looks complex, but I think it is reproducible given the code.

**Strength And Weaknesses:**

Strengths
1. The paper is well written.
2. The studied problem (why DARTS often collapses) is important for the NAS community.
3. The provided explanation is somewhat new.

Weaknesses
1. One of the most significant issues for all recent research in DARTS (not only for this paper) is the limited ability to justify the statements by experiments. Let me explain. The search space of DARTS is relatively small and well studied. The community knows a few tricks (e.g. increasing the depth, or constraining the number of skip-connections) that can easily generate strong architectures in the space. On the opposite, the gap among recognition accuracy of different methods is very small, e.g. 0.1% on CIFAR10 and 0.2-0.3% on ImageNet-1K. This makes it quite difficult to judge whether the improvement indeed comes from the proposed method (or simply an implementation trick).

Going deeper, we know that DARTS underwent a few approximations, e.g. the 1st-order method discards higher-order gradients, and the computation of the inverse Hessian matrix is **largely** approximated -- see the analysis in [Zela et al., ICLR20] and [Bi et al., arXiv19]. The question is that, the proposed method used another strong approximation to alleviate the issue caused by an existing approximation. Integrating it with the above concerns, I cannot believe that the marginal improvement (the authors may argue that the improvement on CIFAR100 is relatively large, but this is because the CIFAR100 is not well tuned by other methods) indeed comes from the proposed method.

To further validate the effectiveness, the authors may try to (i) use a more complex search space (e.g. the spaces constructed in [Bi et al., arXiv19] -- although the authors investigated the spaces offered by [Zela et al., ICLR20], but these spaces are actually even simpler than the original DARTS space) so that the design tricks have weaker impacts, or (ii) report search results with more epochs (e.g. 200 or more), since the collapse issue happens more frequently.

It is also good to discuss the relationship to other works that discussed the stability of DARTS, e.g. DARTS+ [Liang et al., arXiv], XNAS [Nayman et al., NeurIPS19], Fair DARTS [Chu et al., ECCV20], GOLD-NAS [Bi et al., arXiv20].

**Summary Of The Paper:**

This paper presents a differentiable NAS approach based on DARTS. The main discovery is that "DARTS suffers from a specific structural flaw" so that it "gives an unfair advantage to layers closer to the output". To solve this issue, a new regularization method is proposed by "harmonizing operation selection via aligning gradients of layers". Experiments look good on a few benchmarks.

**Summary Of The Review:**

I think the efforts in solving the collapse problem are good and the explanation is interesting, but the empirical results are insufficient to fully justify the statements.

---

> ### Author Response · Authors · 2022-11-18
> **Response to Reviewer iL42 (1/2)**
>
> We would like to thank the reviewer iL42 for the comments, feedback, and suggestions. Below, we provide responses to each comment and pointers to the revised parts in the paper.
>
> **The source of low layer alignment and improvements of the proposed method** We thank the reviewer for their feedback. In the revised version of the paper, we highlight and identify the root cause of the low layer alignment problem, and by extension, the issue of performance collapse. In order to further justify the proposed method, we added a new section to the paper in Appendix-A2 dedicated to discussing the main cause of low layer alignment. We first investigate the approximation of the architecture gradients as one source, in which we empirically show that the issue of low layer alignment still persists after using more accurate estimation methods of the architecture gradients. Then, we provide a theoretical analysis that connects the issue of low layer alignment to the iterative estimation theory [10], which also implicitly links our work to the findings of DARTS-PT [9].  This further enriches the theoretical aspect of our work, while also further distancing our proposed method from the approximation issue rightfully mentioned by the reviewer. On a final note, we would like to note that even if our method is somehow implicitly solving the issues caused by the approximation of the architecture gradients in the first-order DARTS method, it is still providing valuable insights as to how this approximation is causing the performance collapse, and how to resolve it with a method that does not require the costly second-ordered methods such as second-order DARTS, iDARTS [12], and Amended-DARTS [11], all of which are significantly more expensive than $\Lambda$-DARTS due to complex calculations related to the implicit function theorem. Furthermore, we note that all of the aforementioned second-order methods perform poorly compared to our method, which further proves that the theoretical analysis provided in this paper is at least partially orthogonal to the issue of approximation in the first-order DARTS.
>
> **Further experiments** We thank the reviewer for this comment.  We have performed the suggested experiment - namely performing the search on the DARTS search space for 200 epochs - which the dear reviewer can see in Appendix-A6. $\Lambda$-DARTS successfully discovers well performing architectures when performing the search for 200 epochs, at a lower cost compared to Amended-DARTS, which was our main baseline here. Furthermore, we note that we have added the results of performing the search for 400 epochs on the NAS-Bench-201 search space in Figure4c, which further proves the robustness of $\Lambda$-DARTS when searching for a larger number of epochs. We note that to the best of our knowledge, our work currently yields state-of-the-art results for this setting on the DARTS search space and the NAS-Bench-201 search space. We have also provided the results of combining our proposed method with two variants of DARTS (namely P-DARTS and SDARTS) in Appendix-A5, which shows the orthogonality between our proposed method and these methods. If the dear reviewer deems it necessary, we can also perform the search on the larger search space proposed by [11], which for now we omitted from performing due to a lack of resources. We also note that we are currently in the process of performing the search on the ImageNet dataset, which we failed to add to the revisioned version due to a lack of resources. Unfortunately, with our limited computational resources, the experiments are not finished yet, but we expect to be able to collect and report the results in the next few days.
>
> **Relationship to prior works** We thank the reviewer for their feedback. In the revised version of the paper, in Appendix-A3, we have provided a thorough investigation in the relationship between the proposed method and some of the notable prior works, namely $\beta$-DARTS [7], DARTS- [4], PC-DARTS [6], DrNAS [8], and SNAS [2]. We also point the reviewer to Appendix-A5 for results that show the orthogonality between our proposed method and P-DARTS [1] and SDARTS [13], and Appendix-A2 which ties some of our findings to the theoretical analysis provided in DARTS-PT.

---

> ### Author Response · Authors · 2022-11-18
> **Response to Reviewer iL42 (2/2)**
>
> ----------------------------
> [1] Chen X, Xie L, Wu J, Tian Q. Progressive darts: Bridging the optimization gap for nas in the wild. International Journal of Computer Vision. 2021 Mar;129(3):638-55.
>
> [2] Xie S, Zheng H, Liu C, Lin L. SNAS: stochastic neural architecture search. In International Conference on Learning Representations 2018 Sep 27.
>
> [3] Cai H, Zhu L, Han S. ProxylessNAS: Direct Neural Architecture Search on Target Task and Hardware. In International Conference on Learning Representations 2018 Sep 27.
>
> [4] Chu X, Wang X, Zhang B, Lu S, Wei X, Yan J. DARTS-: Robustly Stepping out of Performance Collapse Without Indicators. In International Conference on Learning Representations 2020 Sep 28.
>
> [5] Zela A, Elsken T, Saikia T, Marrakchi Y, Brox T, Hutter F. Understanding and Robustifying Differentiable Architecture Search. In International Conference on Learning Representations 2019 Sep 25.
>
> [6] Xu Y, Xie L, Zhang X, Chen X, Qi GJ, Tian Q, Xiong H. PC-DARTS: Partial Channel Connections for Memory-Efficient Architecture Search. In International Conference on Learning Representations 2019 Sep 25.
>
> [7] Ye P, Li B, Li Y, Chen T, Fan J, Ouyang W. $\beta $-DARTS: Beta-Decay Regularization for Differentiable Architecture Search. arXiv preprint arXiv:2203.01665. 2022 Mar 3.
>
> [8] Chen X, Wang R, Cheng M, Tang X, Hsieh CJ. DrNAS: Dirichlet Neural Architecture Search. In International Conference on Learning Representations 2020 Sep 28.
>
> [9] Wang R, Cheng M, Chen X, Tang X, Hsieh CJ. Rethinking Architecture Selection in Differentiable NAS. In International Conference on Learning Representations 2020 Sep 28.
>
> [10] Greff K, Srivastava RK, Schmidhuber J. Highway and residual networks learn unrolled iterative estimation. arXiv preprint arXiv:1612.07771. 2016 Dec 22.
>
> [11] Bi K, Hu C, Xie L, Chen X, Wei L, Tian Q. Stabilizing darts with amended gradient estimation on architectural parameters. arXiv preprint arXiv:1910.11831. 2019 Oct 25.
>
> [12] Zhang M, Su SW, Pan S, Chang X, Abbasnejad EM, Haffari R. idarts: Differentiable architecture search with stochastic implicit gradients. In International Conference on Machine Learning 2021 Jul 1 (pp. 12557-12566). PMLR.
>
> [13] Chen X, Hsieh CJ. Stabilizing differentiable architecture search via perturbation-based regularization. In International conference on machine learning 2020 Nov 21 (pp. 1554-1565). PMLR.

---

### Official Review · Reviewer_iyeQ · 2022-10-26

**Confidence:** 3
**Correctness:** 4
**Technical Novelty And Significance:** 3
**Empirical Novelty And Significance:** 3
**Recommendation:** 8

**Clarity, Quality, Novelty And Reproducibility:**

The presentation is clear and the derivation of the optimization steps on the modified bi-level darts is solid.


**Strength And Weaknesses:**

Strength:
The observation and analysis of performance collapse in darts is from a novel perspective.
And the regularization terms used in bi-level darts are solid and useful.

Weaknesses:
1. In table 3, CIFAR10 and SVHN dataset on S1 search space, the performance of the proposed method is worse than previous methods.  Seem there are no detailed analyses of this phenomenon.  It is better to show the visual comparisons of searched architectures between previous methods and the proposed method (if possible), and compare the values of layer alignment metric that these previous methods can obtain, on S1 search space (if possible).
2. What's the effect when putting the proposed regularization term for upper-level optimization on  $\alpha$  in bi-level Darts?  The proposed metric is currently only used in lower-level optimization for $w$.



**Summary Of The Paper:**

This paper proposes two new regularization terms to prevent performance collapse by harmonizing operation selection via aligning gradients of layers. The extensive experiments demonstrate its superiority.

**Summary Of The Review:**

I give 6 due to several concerns in the weakness part. I may change the score according to the author's response.

---

> ### Author Response · Authors · 2022-11-18
> **Response to Reviewer iyeQ**
>
> We would like to thank the reviewer iyeQ for the comments, feedback, and suggestions.  We also appreciate that the reviewer is willing to increase their score based on our response. Below, we provide responses to each comment and pointers to the revised parts in the paper.
>
> **S1 results in table 3** As noted by the reviewer, our method appears to be performing slightly worse than the baselines on the S1 search space on the CIFAR-10 and SVHN datasets. We attribute this issue to the unnecessarily deep architectures discovered by our proposed method in the S1 search space, as can be seen in figures 10 and 18, which is not the case in SDARTS [1] (the baseline performing better than $\Lambda$-DARTS in the aforementioned search spaces). We attribute this observation to several reasons. The first important point to note is that our method appears to be orthogonal to the proposed method of SDARTS. This is due to the fact that we are not making any attempts at addressing the Hessian regularization technique, which appears to be improving the performance result of DARTS. Furthermore, we draw the attention of the reviewer to Appendix-A5.2, which clearly shows an orthogonal relationship between our proposed method and SDARTS. So it is entirely possible that our poor performance on S1 on two datasets can be attributed to this issue, and further improvements can be achieved by combining the two methods. The second important point is the overall structure of the S1 search space. We note that S1 is deliberately designed in a way to be a difficult search space, as evident by the results, with different candidate operations on each edge. Specifically, the candidate operations are selected based on eliminating the least important operations (according to their corresponding value of $p$) iteratively after running DARTS on its search space twice [2]. This procedure resulted in a search space that contains some edges with only non-parametric operations. As discussed in Appendix-A2, we attribute the problem of low layer alignment to the convolution operations. Interestingly, our experiments show that the search model on S1 generally starts with an order of magnitude larger layer alignment compared to other search spaces (about 0.02 on S1, as opposed to about 0.001 on S2, S3, and S4), which indicates that our proposed method may be having limited utility in this case. Thirdly, we note that the hyper parameters were not optimized in the reduced search spaces, which is sure to cause some drop in performance in all cases. This is due to the simple fact that we did not have enough resources to perform hyper parameter search for these search spaces, and the bulk of our processing power was dedicated to the DARTS search space, which has a higher cost compared to the reduced search spaces, as expected.
>
> **Performing regularization on the outer optimization problem** We thank the reviewer for such an interesting question. We have added a dedicated section in Appendix-A4 to address this question, which shows performing the regularization on the outer objective does not perform as well as regularizing the inner objective. We have provided a thorough analysis for this observation as well, which points to two factors that may have caused this issue: 1) lack of complexity of the model w.r.t. the architecture parameters when dealing with the regularization objective, 2) the root cause of low layer alignment, as shown in Appendix-A2. So in Appendix-A4 we show that the problem of low layer alignment is inherently related to $\omega$, and cannot be addressed by optimizing $\alpha$ properly, which results in poor performance of the setting proposed by the reviewer.
>
>
> ------------
>
> [1] Chen X, Hsieh CJ. Stabilizing differentiable architecture search via perturbation-based regularization. In International conference on machine learning 2020 Nov 21 (pp. 1554-1565). PMLR.
>
> [2] Zela A, Elsken T, Saikia T, Marrakchi Y, Brox T, Hutter F. Understanding and Robustifying Differentiable Architecture Search. In International Conference on Learning Representations 2019 Sep 25.

---

> > ### Comment · Reviewer_iyeQ · 2022-12-04
> > **Concerns have been addressed.**
> >
> > Thanks for the detailed responses.  I think the novelty of the proposed method is good. The comparisons and the analyses are solid now.
> > I changed the score to 8.

---

### Official Review · Reviewer_2qFj · 2022-10-26

**Confidence:** 4
**Clarity, Quality, Novelty And Reproducibility:** This paper is presented well.
**Correctness:** 3
**Technical Novelty And Significance:** 3
**Empirical Novelty And Significance:** 2
**Recommendation:** 6

**Strength And Weaknesses:**

Strengths:
1. This paper introduces a function named layer alignment to measure the correlation between the gradient of layers in DARTS, theoretical and empirical results show that this function might be a good indicator of the performance collapse situation. This could be a benefit for future works to better understand DARTS.

2. The proposed regularization terms are simple to adopt, and they can alleviate the performance collapse problem according to the experimental results.

Weaknesses:
1. The performance collapse problem in this paper is not novel, which has been widely discussed and solved by many works since 2019.

2. The performance improvements are marginal. On NAS-Bench-201, the performance of the proposed method is the same as previous work $\beta$-DARTS. In Table 2, the improvements on CIFAR-10 and CIFAR-100 are marginal, the results on ImageNet are not state-of-the-art.

Questions:
1. Can the authors provide the search cost of the proposed method? Besides, extensions of the method to other DARTS variants (e.g., some variants that can directly search on large-scale datasets) are preferred to better validate the method.

2. In Figure 4 (c), why the performance drops after 100 epochs? Does the proposed method just delay the performance collapse process?

**Summary Of The Paper:**

This paper proposes a new differentiable architecture search (DARTS) variant to alleviate the performance collapse problem in DARTS. The authors first give theoretical and empirical analysis of the convergence of DARTS, which shows that the weight-sharing framework causes the undesirable skip-connection dominance problem in posterior layers. Based on the analysis, the paper introduces two regularization terms to alleviate the performance collapse problem via aligning gradients of layers.


**Summary Of The Review:**

Overall, I think this paper is a borderline paper. My major concern of this paper is the limited contribution of novelty as well as the performance to the community.

---

> ### Author Response · Authors · 2022-11-18
> **Response to Reviewer 2qFj (1/3)**
>
> We would like to thank the reviewer 2qFj for the comments, feedback, and suggestions. Below, we provide responses to each comment and pointers to the revised parts in the paper.
>
> **Novelty of performance collapse** The problem of performance collapse, is indeed not new and we do not claim any novelty for discovering the issue or introducing the problem. However, the performance collapse in DARTS is still an active issue in the community.
> In our paper, we propose a new approach, which involves a never before done theoretical analysis of the convergence of DARTS, and a new criterion (layer alignment), which we firmly establish has a high correlation with the performance of the discovered architecture. In the revisioned version, we have provided the relationship between our work and some of the prior works in Appendix-A3, which further contextualizes it with respect to other attempts at solving the performance collapse issue. Furthermore, we provide the root cause of low layer alignment in Appendix-A2, which involves a novel approach in establishing a connection between performance collapse in DARTS and the iterative estimation theory in ResNets, tying our method to the findings of DARTS-PT [9]. We also show the orthogonality of our work to some of the variants of DARTS in Appendix-A5, which further supports the novelty of our approach.
>
> **Significance of improvements** We thank the reviewer for this comment. For the NAS-Bench-201 search space, we note that the architecture discovered by our method and $\beta$-DARTS are nearly optimal. Therefore, there was not much room for growth left on this search space. The difference between the optimal architectures on different datasets in the NAS-Bench-201 search space is one operation out of six (3x3 convolutions replaced by 1x1 convolutions and vice versa). So it is practically impossible to get anything better than marginal improvements on this search space. Furthermore, about the ImageNet results, we note that the best-performing model is DARTS- [4], which performs the search and evaluation on a different backbone architecture. About the other methods that perform slightly better than ours (namely, $\beta$-DARTS [7] and DrNAS [8]) we note that their marginal improvement can be attributed to three major differences with our experiment setup: 1) difference in evaluation hyper parameters, 2) the lack of adherence to the mobile setting (<600M FLOPs complexity in the evaluation model), 3) searching on the ImageNet directly. For the first case, we had to reduce the batch size significantly during evaluation due to a lack of computational resources, which we imagine has had a negative impact on the performance.  For the second case, we note that according to our experience, architectures with more than 5.2 million parameters usually violate the mobile setting, which is very important when considering the practical applications of DARTS in the real world. On the other hand, this over complexity is sure to help the performance of the discovered architecture on a complex dataset such as ImageNet, which is the source of the marginally better performance of $\beta$-DARTS on ImageNet compared to $\Lambda$-DARTS. For the third case, we note that both DARTS- and DrNAS have performed the search on ImageNet, which makes a comparison between our method and theirs somewhat unfair. So we are currently in the process of performing the search on the ImageNet dataset. Unfortunately, with our limited computational resources, the experiments are not finished yet, but we expect to be able to collect and report the results in the next few days.
>
> **Search cost** We thank the reviewer for this fine point. In the revised version of the paper, we have provided a thorough analysis of the search cost of our method along with a low-cost variant in Appendix-A7. The analysis contains a thorough comparison with the current state-of-the-art methods both in terms of performance and the cost of search on the CIFAR-10 dataset. We have provided results for the NAS-Bench-201 search space for the low-cost variant in order to show the very limited drop in performance in this case, which proves that the method still performs better than most baselines despite lower accuracy in the estimation of the regularization term gradient.

---

> ### Author Response · Authors · 2022-11-18
> **Response to Reviewer 2qFj (2/3)**
>
> **Extension to other variants of DARTS** We thank the reviewer for their comment and suggestion. In the revisioned version of the paper, we have provided the result of combining our proposed method with two orthogonal cases (namely P-DARTS [1] and SDARTS [10]) in Appendix-A5. In the case of P-DARTS, we showed that our method completely eliminates the need for strong prior assumptions in the search phase, while in SDARTS we eliminated the need for costly adversarial training (as done in SDARTS-ADV), improving the results of SDARTS-RS using the proposed method. P-DARTS is a low-cost variant of DARTS, and combining the proposed method with it has further improved the search cost of $\Lambda$-DARTS, as pointed out by the reviewer. Furthermore, we have provided a thorough investigation of the relationship between our proposed method and some of the baselines (namely DARTS-, PC-DARTS, $\beta$-DARTS, DrNAS, and SNAS [2]) in Appendix-A3 to further contextualize our work and make an assessment about the importance of the theoretical analysis provided in the paper easier. Finally, we have also investigated the root cause of the issue of low layer alignment in Appendix-A2, which further enriches the theoretical aspects of the paper.
>
> **Delaying performance collapse vs. solving performance collapse** We thank the reviewer for their question. As pointed out by the reviewer, the performance of the discovered architecture drops slightly after 100 epochs. But we note that this drop is extremely negligible (less than 0.5%, as noted in paragraph 3 of Section-5.4), which corresponds to a difference in the selected operation on one edge out of six edges, and we attribute it to other factors such as the value of lambda (which was not optimized for larger epochs) and also overfitting. In order to better visualize this fact, we have rescaled the y-axis of all the plots in Figure 4 (to the range 80% - 92%), and also provided another version of figure 4c in figure-8 in the appendix with a baseline (corresponding to the performance of DARTS) for further clarification. Furthermore, in order to provide more evidence for the claim that our method does indeed solve the performance collapse problem, we have also provided the results for 400 epochs of search, which performs better than the experiments with 150 and 200 epochs of search, with a negligible drop in performance compared to the experiment with 100 epochs of search (less than 0.2%) and zero variance. We also draw the attention of the dear reviewer to figure 3 in the paper, which clearly shows signs of convergence in $\Lambda$-DARTS after about 40 epochs of search. We have observed similar effects in all experiments with a larger number of search epochs, which empirically proves that our proposed method is not delaying the performance collapse, but rather facilitates convergence to high-quality architectures. We can provide the convergence plots for the higher number of search epochs, along with the performance of the discovered architectures for all of the datasets for the proposed method on more than 100 epochs of search if the dear reviewer deems it necessary. Finally, we have also added Appendix-A6 to the revisioned version to further show the robustness of our method when searching for longer epochs on the more complex search space of DARTS, which further proves that $\Lambda$-DARTS is successfully alleviating performance collapse, and not delaying it.

---

> ### Author Response · Authors · 2022-11-18
> **Response to Reviewer 2qFj (3/3)**
>
> ----------------------
> [1] Chen X, Xie L, Wu J, Tian Q. Progressive darts: Bridging the optimization gap for nas in the wild. International Journal of Computer Vision. 2021 Mar;129(3):638-55.
>
> [2] Xie S, Zheng H, Liu C, Lin L. SNAS: stochastic neural architecture search. In International Conference on Learning Representations 2018 Sep 27.
>
> [3] Cai H, Zhu L, Han S. ProxylessNAS: Direct Neural Architecture Search on Target Task and Hardware. In International Conference on Learning Representations 2018 Sep 27.
>
> [4] Chu X, Wang X, Zhang B, Lu S, Wei X, Yan J. DARTS-: Robustly Stepping out of Performance Collapse Without Indicators. In International Conference on Learning Representations 2020 Sep 28.
>
> [5] Zela A, Elsken T, Saikia T, Marrakchi Y, Brox T, Hutter F. Understanding and Robustifying Differentiable Architecture Search. In International Conference on Learning Representations 2019 Sep 25.
>
> [6] Xu Y, Xie L, Zhang X, Chen X, Qi GJ, Tian Q, Xiong H. PC-DARTS: Partial Channel Connections for Memory-Efficient Architecture Search. In International Conference on Learning Representations 2019 Sep 25.
>
> [7] Ye P, Li B, Li Y, Chen T, Fan J, Ouyang W. $\beta $-DARTS: Beta-Decay Regularization for Differentiable Architecture Search. arXiv preprint arXiv:2203.01665. 2022 Mar 3.
>
> [8] Chen X, Wang R, Cheng M, Tang X, Hsieh CJ. DrNAS: Dirichlet Neural Architecture Search. In International Conference on Learning Representations 2020 Sep 28.
>
> [9] Wang R, Cheng M, Chen X, Tang X, Hsieh CJ. Rethinking Architecture Selection in Differentiable NAS. In International Conference on Learning Representations 2020 Sep 28.
>
> [10] Chen X, Hsieh CJ. Stabilizing differentiable architecture search via perturbation-based regularization. In International conference on machine learning 2020 Nov 21 (pp. 1554-1565). PMLR.

---

### Official Review · Reviewer_hXod · 2022-10-26

**Confidence:** 3
**Correctness:** 3
**Technical Novelty And Significance:** 3
**Empirical Novelty And Significance:** 3
**Recommendation:** 6

**Clarity, Quality, Novelty And Reproducibility:**

Clarity: good.

Quality: good.

Novelty: good.

Reproducibility: No experimental setup and implementation in the paper.



**Strength And Weaknesses:**

Strength:
1. It has a good starting point and the results of proposed methods can solve the mentioned issues to some extent.

2. The writing is good and easy to understand. A good analysis of lambda w.r.t. the DARTS performance collapse is provided. Low value for lambda means the optimal architecture corresponding to each layer varies wildly. This work seems to be the first to investigate DARTS from the weight-sharing framework and convergence conditions.

Weakness:

For DARTS tasks, the search cost (GPU days) is a crucial evaluation. But the authors did not mention it in the experimental results.

Although the test accuracy is better than other methods, the number of parameters seems to be larger than most of the previous methods. Since the parameters are one of the factors in Table 2, I wonder how to explain this factor.

Tables 1 and 2 reports test accuracy, while Table 3 reports test error rate. I wonder if there is any cherry pick?

Also the performance of ImageNet is not reported in these tables. Is is known that NAS runs slow and how best to scale NAS to run on large datasets such as ImageNet is crucial for real-world applications. And as I know many DARTS methods can only search on toy datasets such as CIFAR-10. Can you report if your NAS method can be trained on large-scale dataset?

**Summary Of The Paper:**

This paper finds the main reason behind performance collapse in DARTS. A new algorithm named  lambda-DARTS is proposed, which is able to generate better performance compared with other methods. Specifically, two new regularization terms are proposed to prevent performance collapse by harmonizing operation selection via aligning gradients of layers. The lambda denotes the dubbed layer alignment, which is a measure to show the correlation between the gradient of each layer corresponding to the architecture parameters.

**Summary Of The Review:**

Overall, I think it is a good paper with a good starting point and clear writing. I mentioned 2 doubts in the weakness part.

Also, there are many existing works on the performance collapse of DARTS. For example, the MS-DARTS which leverages the mean shift to prevent performance collapse at the final discretization step of DARTS.

---

> ### Author Response · Authors · 2022-11-18
> **Response to Reviewer hXod (1/2)**
>
> We would like to thank the reviewer hXod for the comments, feedback, and suggestions. Below, we provide responses to each comment and pointers to the revised parts in the paper.
>
> **Search cost** We thank the reviewer for this fine point. We have provided a thorough analysis of the search cost of our method along with a variant of the proposed method with a lower cost in Appendix-A7. The analysis also contains a comparison with other state-of-the-art methods both in terms of cost and performance. We have provided results for the NAS-Bench-201 search space for the low-cost variant in order to show the very limited drop in performance in this case, which proves that the method still performs better than most baselines despite lower accuracy in the estimation of the regularization term gradient.
>
> **Number of parameters** Thanks for the question.  This is indeed a valid point.  First, we would like to highlight that the source of parametric complexity in most DARTS-related search spaces is two-fold: 1) the number of convolution operations, and 2) the kernel size of the convolution operations. According to our experience, and also confirmed by [1], DARTS search spaces are usually designed in a way that the best-performing architectures are also the ones with the largest possible number of convolution operations, with a single skip-connection to account for vanishing gradients. This means that if we were to optimize $\alpha$ according to the validation performance, most optimal architectures would have a large number of convolution operations in them. But interestingly, the second source of complexity - namely kernel size - doesn’t have a huge impact on performance according to our experience, and only helps with performance in the ImageNet dataset. This, of course, is one of the reasons behind 3x3 convolutions becoming the standard in most notable convolutional neural networks. We note that most of the architectures discovered by our method contain a very small number of 5x5 convolutions, and as you can see in figures 8 and 9, this is also the case for the best-performing architectures discovered by our method. But this is not the case in some of the baselines, namely DrNAS [8] and $\beta$-DARTS [7], which have a larger number of parameters compared to $\Lambda$-DARTS (DrNAS with 4 million and $\beta$-DARTS with an average of 3.8 million, compared to $\Lambda$-DARTS with an average of 3.6 million). This, in turn, results in their slightly better performance on ImageNet, which also doesn’t abide by the mobile setting of ImageNet. To the best of our knowledge, the parametric complexity of the DARTS method is only addressed in special variants such as SNAS and ProxylessNAS [2, 3], which are not directly applicable to our method. So addressing this issue was only possible through dedicating some of our work to formulating the concept of parametric complexity in DARTS, which was not the main subject of our work. But given the importance of this issue, we concur that it needs to be addressed in future works.
>
> **Test error in Table 3 vs accuracy in Tables 1 and 2** Thanks for the question.  We were simply following the lead of one of the baselines (DARTS- [4]) in using the mentioned format. In [4], the performance of the model for the NAS-Bench-201 search space and the DARTS search space are reported based on accuracy in Tables 2 and 3 (​​https://arxiv.org/pdf/2009.01027.pdf#page=6), while the performance of the model for the reduced search spaces is reported based on error rate (100 - accuracy) in Table 7 (https://arxiv.org/pdf/2009.01027.pdf#page=7). Accordingly, we strictly followed the setting of [5] in performing the experiments on the reduced search space, which involves performing the search four times for each setting and reporting the error rate of the best-performing architecture, as reported in Table 1 in [5] (https://arxiv.org/pdf/1909.09656.pdf#page=7). Of course, if the reviewer deems it necessary, we can instead report the accuracy for all experiments, which is simply 1-errors reported in Table 3 of our paper.
>
> **ImageNet results** We thank the reviewer for the comment. We acknowledge the importance of your suggestion on including ImageNet results.  . As can be inferred from Appendix-A7, our method does not have a significant overhead compared to DARTS and is capable of performing the search on ImageNet as well. Currently, we are in the process of providing the ImageNet search results. But unfortunately, we were not able to finish the experiment before the deadline of the rebuttal phase due to a lack of resources and the large number of experiments we had to do for the rebuttal phase. We expect to have the results in the next few days and will reply to this comment as soon as the experiments are done.

---

> ### Author Response · Authors · 2022-11-18
> **Response to Reviewer hXod (2/2)**
>
> **Existing works and novelty** As pointed out by some of the reviewers, the problem of performance collapse is basically as old as NAS itself. So of course, we do not claim any novelty for discovering the issue, but we note the novelty of the approach, which involves a new theoretical analysis of the convergence of DARTS and proposing a novel criterion (layer alignment), which we show has a high correlation with the performance of the discovered architecture. In the revisioned version of the paper, we tried to further contextualize our proposed method by pointing out the relationship between our findings and some of the prior works (namely, DARTS- [4], PC-DARTS [6], $\beta$-DARTS [7], DrNAS [8], and SNAS [2]) in Appendix-A3. Furthermore, we also tried to discover the root cause of the problem of low layer alignment in DARTS in Appendix-A2, which we believe further enriches our work and provides new paths for further research, while also tying our method to the findings of DARTS-PT [9]. We also note that we have successfully shown the orthogonality of our method with two other variants of DARTS (namely P-DARTS [1] and SDARTS [10]) in Appendix-A5, which further supports the novelty of our work.
>
> -----------------------
> [1] Chen X, Xie L, Wu J, Tian Q. Progressive darts: Bridging the optimization gap for nas in the wild. International Journal of Computer Vision. 2021 Mar;129(3):638-55.
>
> [2] Xie S, Zheng H, Liu C, Lin L. SNAS: stochastic neural architecture search. In International Conference on Learning Representations 2018 Sep 27.
>
> [3] Cai H, Zhu L, Han S. ProxylessNAS: Direct Neural Architecture Search on Target Task and Hardware. In International Conference on Learning Representations 2018 Sep 27.
>
> [4] Chu X, Wang X, Zhang B, Lu S, Wei X, Yan J. DARTS-: Robustly Stepping out of Performance Collapse Without Indicators. In International Conference on Learning Representations 2020 Sep 28.
>
> [5] Zela A, Elsken T, Saikia T, Marrakchi Y, Brox T, Hutter F. Understanding and Robustifying Differentiable Architecture Search. In International Conference on Learning Representations 2019 Sep 25.
>
> [6] Xu Y, Xie L, Zhang X, Chen X, Qi GJ, Tian Q, Xiong H. PC-DARTS: Partial Channel Connections for Memory-Efficient Architecture Search. In International Conference on Learning Representations 2019 Sep 25.
>
> [7] Ye P, Li B, Li Y, Chen T, Fan J, Ouyang W. $\beta $-DARTS: Beta-Decay Regularization for Differentiable Architecture Search. arXiv preprint arXiv:2203.01665. 2022 Mar 3.
>
> [8] Chen X, Wang R, Cheng M, Tang X, Hsieh CJ. DrNAS: Dirichlet Neural Architecture Search. In International Conference on Learning Representations 2020 Sep 28.
>
> [9] Wang R, Cheng M, Chen X, Tang X, Hsieh CJ. Rethinking Architecture Selection in Differentiable NAS. In International Conference on Learning Representations 2020 Sep 28.
>
> [10] Chen X, Hsieh CJ. Stabilizing differentiable architecture search via perturbation-based regularization. In International conference on machine learning 2020 Nov 21 (pp. 1554-1565). PMLR.

---

### Author Response · Authors · 2022-11-18
**General Response**

We thank all the reviewers for their feedback. We have revised the paper to address questions and comments. The changes are summarized below.

* We added the results of the search for 400 epochs to Figure-4c to show that the performance collapse is prevented even in the most severe case, and not simply delayed.

* We added analytical and empirical evidence on the root of low layer alignment, and by extension, performance collapse. We investigated two candidates, namely first-order estimation of architecture gradients and the iterative estimation theory in ResNets, as the two main candidates. This investigation also connects our proposed method to the findings of DARTS-PT, further enriching the theoretical aspects of the paper.  [Appendix-A2]

* We explained the relationship between our proposed method, both analytically and empirically. We show that the good performance of $\beta$-DARTS, DARTS-, PC-DARTS, DrNAS, and SNAS is explainable through the analytical framework we provided in this paper. This section further contextualizes our proposed method in the literature. [Appendix-A3]

* We further investigate the possibility of performing the regularization on the outer objective of DARTS instead of the inner objective. Here, we first provide an analytical examination of the validity of such an action, along with experimental results to support the findings. [Appendix-A4]

* We added empirical evidence that suggests the orthogonality of the proposed method with two popular DARTS variants, namely P-DARTS and SDARTS. In the first case, we eliminate the need for strong prior assumptions in the search procedure, as done in P-DARTS. This shows the potential of combining our proposed method with a low-cost variant of DARTS. In the second case, we eliminate the need for costly adversarial training in SDARTS. This shows that our method can help reduce the need for strong regularization on the Hessian during search, while also benefiting from Hessian regularization done through low-cost random perturbations. [Appendix-A5]

* We add an investigation of the performance collapse for longer epochs on the complex search space of DARTS. This section empirically shows that our proposed method successfully alleviates performance collapse even in the most severe cases. We also added Figure-4c with a baseline to this section for better visualization. [Appendix-A6]

*  We added a dedicated discussion to the issue of search cost, where we compare the proposed method to the state-of-the-art methods in terms of performance on the CIFAR-10 dataset on the DARTS search space and the search cost. Furthermore, in this section, we propose a low-cost variant of $\Lambda$-DARTS to reduce the cost of search, which is empirically shown to be effective and cause a limited drop in performance. [Appendix-A7]

We respond to comments and questions from each reviewer in more detail.

Authors of Paper3952.

---

### Author Response · Authors · 2022-11-30
**Lamda-DARTS on ImageNet**

We have run a search and evaluation experiment on the ImageNet dataset, which was requested by reviewer hXod. These results will also provide further experimental evidence for the superiority of our method, which will hopefully address the concerns of reviewers 2qFj and iL42 with respect to this matter.

The search was performed on the setting proposed by PC-DARTS, on a search model with 8 layers (6 normal and 2 reduction cells) and 16 initial channels. Following PC-DARTS and DrNAS, we randomly sampled 10% and 2.5% of the ImageNet dataset for optimizing $\omega$ and $\alpha$, respectively. We used a batch size of 384 (significantly lower than the batch size of 1024 used by PC-DARTS and DrNAS, due to limited memory of our gpu) and a learning rate of 0.5 which is linearly decayed to 0.0. The evaluation was performed with a similar setting, for 250 epochs with a model with 14 layers and 48 initial channels. You can see the results in the following table, and the discovered architectures here: [normal](https://drive.google.com/file/d/1DcK5C2JW7ydPnUxunsBg0XnzBC9uKPGW/view?usp=share_link), [reduction](https://drive.google.com/file/d/1n4mNcCC1FzX7rpS7Mq35lMtuqytXe-w1/view?usp=share_link). We will add these results to the next revision of the paper (we did not update the manuscript, as the deadline for revision has already passed).
|         Method          |   ImageNet Top-1 Acc   |    ImageNet Top-5 Acc   |
|-------------------------|-----------------------:|------------------------:|
|     NASNet-A            |          74.0          |          91.6           |
|     DARTS (2nd)         |          73.3          |          91.3           |
|     SNAS	             |          72.7          |          90.8           |
|     GDAS	             |          74.0          |          91.5           |
|     P-DARTS	             |          75.3          |          92.5           |
|     PC-DARTS            |          75.8          |          92.7           |
|     DrNAS               |          75.8          |          92.7           |
|     SDARTS-ADV          |          74.8          |          92.2           |
|     DOTS                |          75.7          |          92.6           |
|     DARTS+PT            |          74.5          |          92.0           |
|     $\beta$-DARTS       |          75.8          |          92.9           |
|     $\Lambda$-DARDT     |        **75.9**        |        **93.0**         |

We can see that our proposed method improves upon all baselines both in terms of top-1 and top-5 accuracy. We improve upon the performance of DARTS by 2.5% in terms of top-1 accuracy and 1.7% in terms of top-5 accuracy. Furthermore, we beat the best-performing baseline, namely $\beta$-DARTS, by a margin of 0.1% in terms of top-1 accuracy and 0.1% in terms of top-5 accuracy. We also note that DrNAS utilizes a slightly different setting for search, which performs the search on a search model with 14 layers and 48 initial channels, similar to the evaluation model. Despite this advantage, our proposed method beats the performance of DrNAS by a margin of 0.1% on top-1 accuracy and 0.3% on top-5 accuracy. Considering the effect of batch size over the batch normalization layers, we expect the results of both search and evaluation to improve significantly if run on a larger batch size.


We would like to thank all the reviewers again for taking the time to give feedback on our work. We put our best effort to address the comments and answer the questions. We appreciate that some reviewers replied already. For others, we are happy to discuss further for any clarification needed or any point that helps reviewers to consider revising their score.

---

### Decision · Program_Chairs · 2023-01-20

**Decision:**

Accept: poster

**Justification For Why Not Higher Score:**

The underlying problem  of DARTS performance is well discussed in the literature. The achieved improvements are rather small, which is, however, also due to the nature of the datasets and search spaces that are commonly considered in NAS.

**Justification For Why Not Lower Score:**

The proposed approach is well motivated and proposes a solution to a widely discussed problem. Therefore, the paper should be accepted to ICLR.

**Metareview: Summary, Strengths And Weaknesses:**

This paper proposes a variant of DARTS that allows to alleviate the performance collapse in DARTS. The prooposed approach is based
on a theoretic analysis of the convergence of DARTS tha tallows to explain the preference of Skip connections in darts. By aligning gradients of layers using regularization, the observed problem can be addressed.

The addressed problem is an important limitation of DARTS and the proposed regularization is effective and easy to adopt. Thereby, the paper is one of many works that address the problem to define regularizaitons that help DARTS' performance. The underlying problem is well discussed in the literature.
The achieved improvements are also rather small, which is, however, also due to the nature of the datasets and search spaces that are commonly considered in NAS.

While this paper initially received borderline reviews, the most important concerns of reviewers could be addressed during the discussion period.

**Note From Pc:**

if the above contains the word "oral" or "spotlight" please see: "oral" presentation means -> notable-top-5% and "spotlight" means -> notable-top-25%. As stated in our emails, we are disassociating presentation type from AC recommendations